# CaMK4 controls follicular helper T cell expansion and function during normal and autoimmune T-dependent B cell responses

Marc Scherlinger [1,2,3] ✉, Hao Li [1], Wenliang Pan[1], Wei Li[1], Kohei Karino [1], Theodoros Vichos [1], Afroditi Boulougoura[1], Nobuya Yoshida[1], Maria G. Tsokos [1] & George C. Tsokos [1] ✉

Systemic lupus erythematosus (SLE) is an autoimmune disease characterized by dysregulated B cell compartment responsible for the production of auto-antibodies. Here, we show that T cell-specific expression of calcium/calmodulin-dependent protein kinase IV (CaMK4) leads to T follicular helper (T_fh) cells expansion in models of T-dependent immunization and autoimmunity. Mechanistically, CaMK4 controls the T_fh-specific transcription factor B cell lymphoma 6 (*Bcl6*) at the transcriptional level through the cAMP responsive element modulator α (CREMα). In the absence of CaMK4 in T cells, germinal center formation and humoral immunity is impaired in immunized mice, resulting in reduced anti-dsDNA titres, as well as IgG and complement kidney deposition in the lupus-prone B6.*lpr* mouse. In human T_fh cells, CaMK4 inhibition reduced *BCL6* expression and IL-21 secretion ex vivo, resulting in impaired plasmablast formation and IgG production. In patients with SLE, *CAMK4* mRNA levels in T_fh cells correlated with those of *BCL6*. In conclusion, we identify CaMK4/CREMα as a driver of T cell-dependent B cell dysregulation in autoimmunity.

Systemic lupus erythematosus (SLE) is an autoimmune disease characterized by a dysregulated B cell compartment which leads to the emergence of pathogenic autoantibodies and subsequent tissue damage[1]. The standard of care relies on non-targeted immunosuppressive drugs such as corticosteroids and cytotoxic drugs, which increase the risk of infection that nowadays represents the first cause of death in SLE[2,3]. Although recent advances in our understanding of SLE pathogenesis have fostered the development of promising targeted treatments[4], many have failed in phase-3 trials. There is, therefore, a major need to identify new therapeutic targets to treat patients with SLE.

T-cell dysregulation has been consistently associated with lupus pathogenesis. We previously reported that SLE T cells are characterized by an aberrant T cell receptor complex (TCR) signaling, whereby spleen tyrosine kinase (Syk) is recruited in the place of Zap70 and induces aberrant signaling including enhanced and earlier intracellular calcium concentrations following the engagement of the CD3/TCR[5,6]. This increased intracytoplasmic calcium is responsible for the activation of protein kinase C and other calcium-controlled kinases, including the serine/threonine calcium-calmodulin dependent kinase IV (CaMK4)[6]. In patients with SLE, especially in those with active disease, increased T cell calcium activates CaMK4, which skews T cells toward the proinflammatory Th17 compartment and promotes T regulatory (Treg) cell dysfunction[7,8]. Interestingly, lupus-prone mice with a *Camk4* genetic deletion are characterized by a normalized T cell subset distribution together with decreased levels of autoantibodies[9], suggesting the importance of CaMK4 in the humoral (auto-) immune response.

[1]Department of Medicine, Beth Israel Deaconess Medical Center, Boston, MA, USA. [2]Rheumatology department, Strasbourg University Hospital of Hautepierre, Strasbourg, France. [3]Laboratoire d'ImmunoRhumatologie Moléculaire, Institut National de la Santé et de la Recherche Médicale (INSERM) UMR_S 1109, Strasbourg, France. ✉e-mail: marc.scherlinger@chru-strasbourg.fr; gtsokos@bidmc.harvard.edu

Since B cells express low levels of CaMK4[10], we hypothesized that CaMK4 might control humoral response through distinct T cell subsets. T follicular helper (T$_{fh}$) cells are a subset of CD4$^+$ T cells which express the *Bcl6* transcription factor and the CXCR5 chemokine receptor, which allows T$_{fh}$ migration into the secondary lymphoid structures through a gradient of CXCL13[11]. T$_{fh}$ cells accumulate in the T zone of the germinal center and drive B cell affinity maturation and isotype switching, notably through the expression of CD40L and the production of interleukin-21 (IL-21). In murine and human SLE, T$_{fh}$ cells have been shown to be abnormally expanded and activated and to drive autoantibody production and tissue damage[12,13]. The mechanisms that mediate T$_{fh}$ expansion are likely multifactorial and include increased T$_{fh}$ differentiation through the OX40/OX40L axis and platelet activation[14,15], transdifferentiation from regulatory T cells (T$_{reg}$)[16], and impaired T follicular regulatory (T$_{fr}$) cell function[17,18]. In the current study, we investigated the importance of the expression of CaMK4 in T cells in the activation and functional expansion of T$_{fh}$ cells in the setting of immunization with nominal antigens and autoantigens.

Using mice with a T cell-specific genetic deletion of *Camk4*, we showed that CaMK4 is responsible for the expansion of murine T$_{fh}$ cells after immunization and is necessary for the development of a full-blown humoral response. Mechanistically, CaMK4 promotes the expression of *Bcl6* through the activation of the cAMP response element modulator α (CREMα) transcription factor. In the B6.*lpr* lupus-prone mouse, the deletion of *Camk4* in T cells leads to the reduction of circulating and tissue T$_{fh}$ cells, decreased levels of multiple pathogenic B cell subsets, and improvement of biological and clinical hallmarks of the disease. At a translational level, CaMK4 inhibition in primary human T$_{fh}$ cells led to the downregulation of *BCL6* expression, IL-21 production, and B cell-stimulating properties of T$_{fh}$. Overall, this study identifies CaMK4 as a modulator of T-dependent humoral immunity in the normal and autoimmune response.

## Results

### CaMK4 drives T$_{fh}$ expansion through the transcriptional control of *Bcl6* expression

We previously found that murine Treg cells express high levels of CaMK4 compared to B cells[8]. To evaluate CaMK4 expression in various immune compartments, we used fluorescent-activated cell sorting to isolate B (B220$^+$CD3$^-$) cells, T$_{fh}$ (CD3$^+$CD4$^+$CD25$^-$CD44$^+$CXCR5$^+$PD1$^+$) cells and T$_{reg}$ (CD3$^+$CD4$^+$CD25$^+$) cells from the splenocytes of C57Bl/6 mice. Using reverse-transcription quantitative qPCR, we confirmed that Treg cells express high levels of *Camk4* while B cells express minimal amounts (Fig. 1a). Interestingly, T$_{fh}$ cells also express high levels of *Camk4* (Fig. 1a), suggesting its role in T$_{fh}$ biology. To evaluate the importance of *Camk4* expression for T$_{fh}$ cell development, we immunized wild-type C57Bl/6 (designed as WT, *Camk4$^{fl/fl}$*) or T cell-specific *Camk4*-deficient (*Camk4$^{fl/fl}$.dlck-Cre*) mice with sheep red blood cells (SRBC). On day 7 after immunization, WT mice had an expansion of CXCR5$^+$ PD1$^+$ T$_{fh}$ cells within splenocytes (Fig. 1b; Fig. S1A, $p < 0.001$), while those with T cell-specific CaMK4 deficiency did not ($p$ = ns). Conversely, CXCR5$^-$PD1$^+$ T peripheral helper (T$_{ph}$) cells were not affected by T cell-specific *Camk4* deletion (Fig. S1B, C). Impaired expansion of Tfh cells upon SRBC immunization was confirmed using intracellular staining for BCL6 (Fig. 1c and Fig. S1D). Mice with global Camk4 deletion had a similar impaired expansion of CXCR5$^+$ PD1$^+$ T$_{fh}$ after immunization (Fig. S1E, F). Since we previously showed that CaMK4 expression negatively affects Treg cell differentiation and function in the settings of autoimmunity[8], we asked whether this was also true in our model of T cell-dependent immunization. Neither T$_{reg}$,

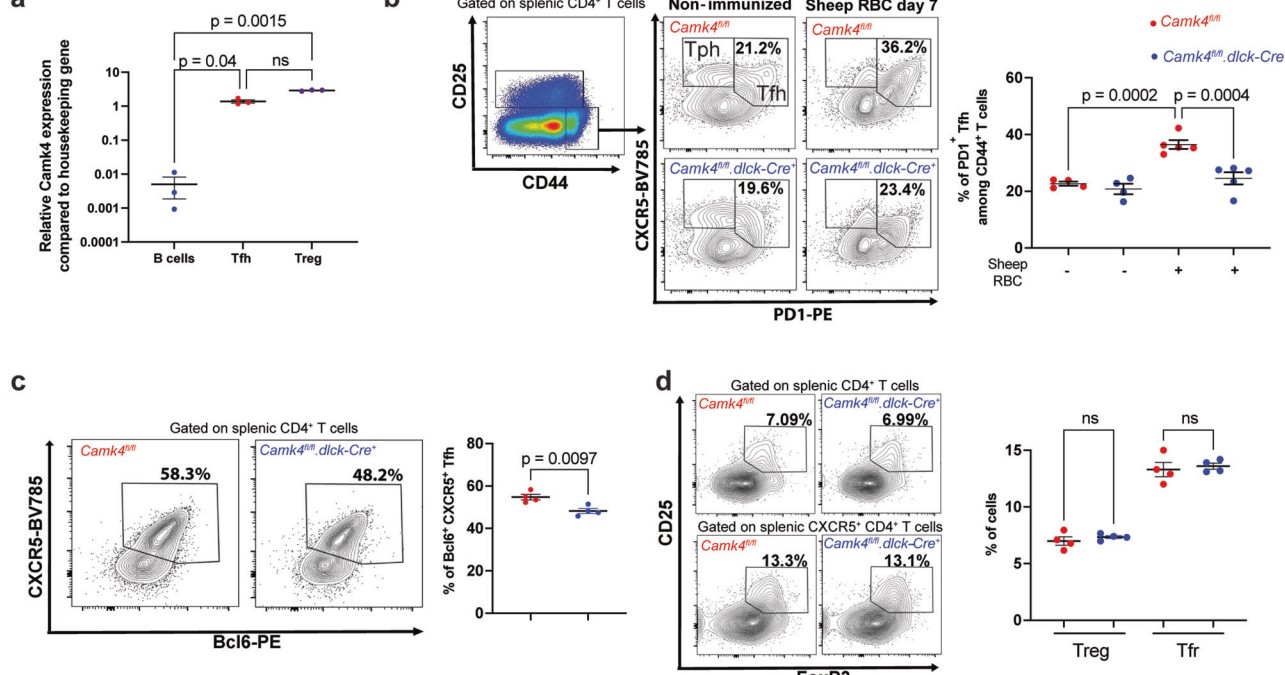

**Fig. 1 | CaMK4 drives T$_{fh}$ expansion during T cell-dependent immunization.**
**a** Splenic B cells (B220$^+$CD3$^-$), T follicular helper (CD3$^+$CD4$^+$ CD44$^+$CD25$^-$CXCR5$^+$PD1$^+$), and T$_{reg}$ (CD3$^+$CD4$^+$CD25$^+$) were sorted using flow cytometry, and RNA was isolated using Trizol. The expression of *Camk4* was evaluated using qPCR ($n$ = 3 mice). **b–d** *Camk4$^{fl/fl}$* (red, $n$ = 5) and *Camk4$^{fl/fl}$.dlck-Cre* (blue, $n$ = 5) mice were immunized by an i.p. injection of sheep red blood cells and subsequently sacrificed after 7 days. Non-immunized mice were used as a control ($n$ = 4 for each genotype). **b** Gating strategy of PD1$^+$ T$_{fh}$ cells and CXCR5$^-$PD1$^+$ T peripheral helper (T$_{ph}$) cells from the spleen of *Camk4$^{fl/fl}$* (blue) and *Camk4$^{fl/fl}$.dlck-Cre* (blue) 7 days after immunization with sheep red blood cells (left panel) and cumulative results (right panel). **c** Gating strategy of splenic CD4$^+$ CXCR5$^+$ Bcl6$^+$ T$_{fh}$ cells (left panel) and cumulative results (right panel). **d** Gating of splenic T$_{reg}$ and T follicular regulatory (T$_{fr}$) cells (left panel) and cumulative results (right panel). Paired one-way ANOVA with Holm−Sidak's correction (**a**) or one-way ANOVA with Holm−Sidak's correction (**b–d**). Bars indicate mean ± s.e.m.

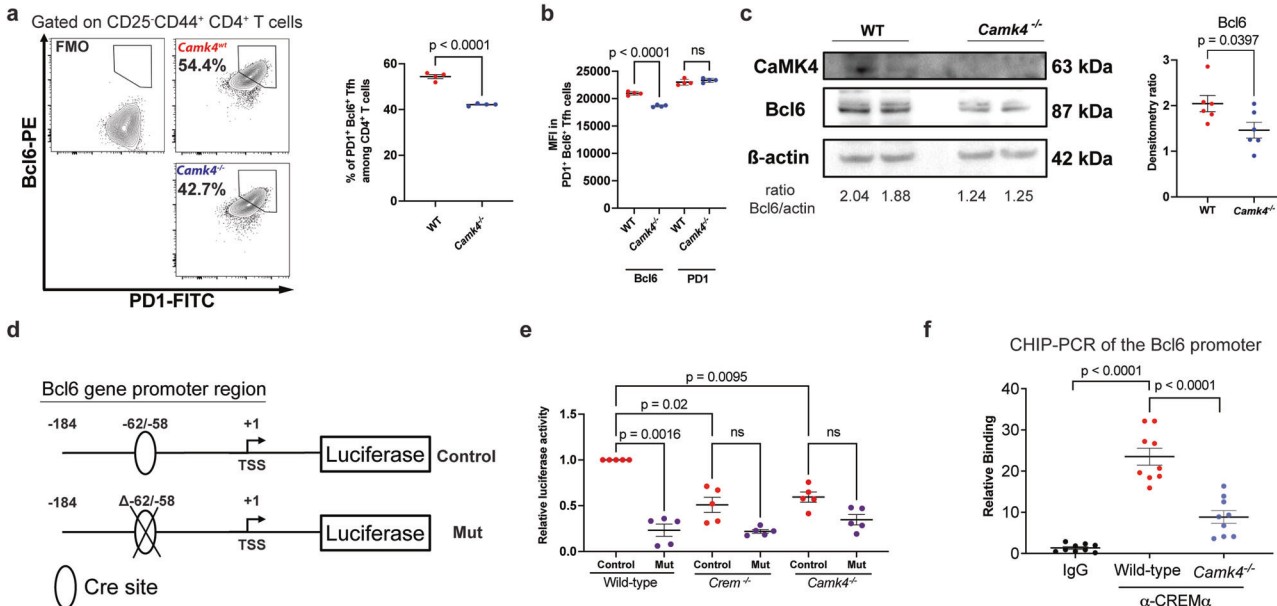

**Fig. 2 | CaMK4 modulates *Bcl6* gene expression through CREMα. a** Wild-type (Wt, red) or *Camk4*[-/-] T cells (blue) CD62L[+] CD4[+] T cells were differentiated in vitro in T_{fh} cells for 3 days (*n* = 4 mice). Gating strategy for differentiation T_{fh} (left panel) and cumulative data (right panel; *n* = 4 mice). **b** Mean fluorescence intensity (MFI) of Bcl6 and PD1 in in vitro differentiated T_{fh} (*n* = 4 mice). **c** Representative western blot of CaMK4, Bcl6, and ß-actin protein expression in vitro differentiated T_{fh} (left panel) and densitometry (right panel; each point represents an independent experiment). **d** Design of the luciferase reporter vector including the full Bcl6 gene promoter (control) or the Bcl6 gene promoter with deletion of the CRE-binding site (Mut). **e** Relative luciferase activity in in vitro differentiated T_{fh} transfected with the control and the mutated luciferase promoter (*n* = 5 mice, conducted in two independent experiments). **f** Chromatin immunoprecipitation (CHIP) was conducted on the DNA of murine iTfh cell using a CREMα or control antibody. The qPCR targeted the *Bcl6* gene promoter region, and relative binding was calculated by comparing the ΔCt between CREMα or control antibody with the input. Three technical replicates are shown for each mouse (*n* = 3 mice per group). Unpaired two-tailed Student's *t*-test (**a**, **c**) one-way ANOVA (**b**) or one-way paired ANOVA (**e**) with Holm−Sidak's correction. Bars indicate mean ± s.e.m.

nor T_{fr} (CD3[+]CD4[+]CD25[+]FoxP3[+]CXCR5[+]) cells were affected by CaMK4 expression at day 7 after SRBC immunization (Fig. S1G, H).

To investigate the molecular mechanism underlying these differences, we differentiated in vitro CD62L[+] CD4[+] T cells to inducible T_{fh} (iT_{fh}) cells (see methods), leading to the expression of Bcl6 and PD1 (Fig. 2a; gating Fig. S2A). CaMK4 deficiency led to a significant downregulation of *Bcl6* (*p* < 0.001) but not PD1 on iT_{fh} (Fig. 2b, c). Conversely, transfection of iTfh cells with a CaMK4-overexpressing vector led to increased Bcl6 expression (Fig. S2B). We and others have previously shown that CaMK4 positively or negatively controls the expression of several genes through the expression of cAMP-responsive element modulation α (CREMα)[7,9]. Indeed, activated CaMK4 binds and phosphorylates (i.e., activates) CREMα, which leads to CREMα relocalization to the nucleus and the modulation of gene transcription. By analyzing the promoter for the murine *Bcl6* gene, we found 2 CRE-binding sites, one of which was highly conserved among mammals (human, mouse, rabbit). We, therefore, hypothesized that CaMK4 may promote *Bcl6* gene expression through CREMα. To address this, we cloned the *Bcl6* gene promoter on a pGL3 *Renilla* luciferase reporter vector and conducted a site-directed mutagenesis to delete the highly conserved CRE-binding site (Mut vector, Fig. 2d). The *Renilla* vector was transfected, along with a control firefly luciferase vector, in cells two days after the initiation of iTfh differentiation and luciferase activity assessed at day 3. The normalized luciferase activity was significantly decreased in the Mutated (Mut) vector compared to the WT vector (*p* < 0.01, Fig. 2e). Furthermore when the reporter vectors were transfected in *Camk4*[-/-] or in *Icer/Crem*[-/-] T cells, the luciferase activity was similarly decreased (Fig. 2e). To confirm these results, we conducted chromatin immunoprecipitation (CHIP−) PCR on iTfh cells (see Methods). DNA pulldown with a CREMα-specific antibody led to significant enrichment of the *Bcl6* gene promoter region (Fig. 2f). Conversely, in *Camk4*[-/-] iTfh, the enrichment was

significantly lower compared to WT iTfh (Fig. 2f), suggesting that CaMK4 expression is instrumental in CREMα binding to the Bcl6 gene promoter region. Overall, our results suggest that CaMK4 is responsible for the expansion of T_{fh} cells during T cell-dependent immunization through the promotion of the *Bcl6* gene expression by CREMα.

## T cell CaMK4 controls germinal center formation and B cell response during immunization

Next, we evaluated T_{fh} differentiation and function in vivo using the murine SRBC immunization model. Splenic germinal center B (GCB; B220[+]CD93[−]CD95[+]GL7[+]) cells were significantly decreased at 7 and 14 days after immunization in mice with T cell-specific *Camk4* knockdown compared to controls (Fig. 3a, b). Before and after immunization, other B cell subpopulations were unremarkable (Fig. S3A−C). Consistently, we observed impaired formation of organized splenic germinal centers in *Camk4*[fl/fl].*dlck*[Cre] mice (Fig. 3c, d). Since germinal center formation is important for positive selection and affinity maturation of IgG-producing B cells[19], we evaluated the humoral response with another model of T cell-dependent immunization, the 4-Hydroxy-3-nitrophenylacetyl chicken γ-globulin (NP-CGG) model (Fig. 4a). After the immunization, the levels of total IgG were similar in *Camk4*[fl/fl].*dlck-Cre* and *Camk4*[fl/fl] mice (Fig. 4b). However, NP-specific GC B cells and plasma cells were significantly decreased in *Camk4*[fl/fl].*dlck-Cre* mice compared to their Camk4-sufficient counterparts (Fig. 4c, d). This was associated with decreased levels of high affinity (NP-7) IgG, but not mixed affinity IgG (Fig. 4e), with a decrease of the anti-NP7/NP44 IgG ratio suggesting an impaired affinity maturation (Fig. 4f). The study of IgG subclasses identified that IgG1 and IgG3 were mainly affected by T cell-specific *Camk4* deletion (Fig S4A−C). Importantly, mice with a global *Camk4* deletion also developed a similarly impaired antibody response to NP-CGG (Fig. S4D). Overall, these results

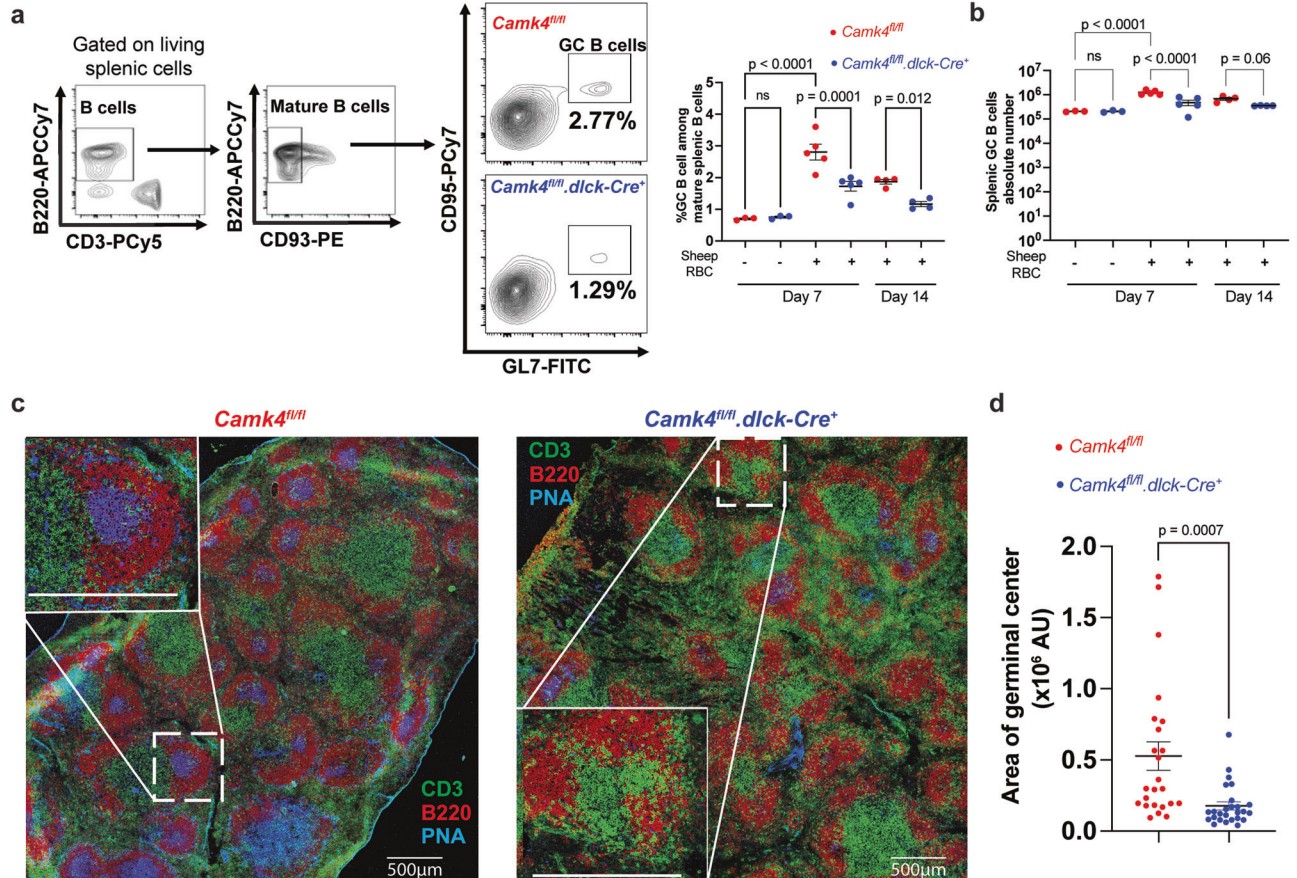

**Fig. 3 | T cell CaMK4 drives germinal center formation during immunization.** *Camk4*^fl/fl^ (red) and *Camk4*^fl/fl^.*dlck-Cre* (blue) mice were immunized with sheep red blood cells and subsequently sacrificed (each point represents an individual mouse). **a** Gating strategy (left panel) and cumulative data (right panel) of splenic germinal center B cells. **b** Absolute number of splenic germinal center B cells. **c** Immunofluorescence of the spleen 7 days after immunization showing T cell (CD3, green), B cells (B220, red), and germinal centers (peanut antigen PNA, blue). **d** The area of germinal centers (PNA staining) was measured in the spleen of *Camk4*^fl/fl^ and *Camk4*^fl/fl^.*dlck-Cre* (n = 3 mice per group). Each dot represents the area of one germinal center (AU arbitrary unit). One-way ANOVA with Holm-Sidak's correction (**a**, **b**) or the two-tailed non-parametric Kolmogorov–Smirnov test (**d**). Bars indicate mean ± s.e.m.

demonstrate that The T-cell expression of CaMK4 positively controls the germinal center formation and the high-affinity humoral response in the setting of T-dependent immunization.

## T cell CaMK4 controls pathogenic B cell development and autoantibody production in autoimmunity

To evaluate the relevance of our findings in autoimmunity, we crossed the B6.*lpr* with the *Camk4*^fl/fl^.*dlck-Cre* mouse to generate a T cell-specific, *Camk4*-deficient lupus-prone mouse. B6.*lpr.Camk4*^fl/fl^ and B6.*lpr.Camk4*^fl/fl^.*dlck-Cre* littermates were sacrificed at 34 weeks of age. B6.*lpr*, which did not express CaMK4 in T cells, was characterized by decreased levels of PD1⁺ T_fh cells in the spleen, cervical lymph nodes (cLNs) and the blood (gating Fig. S5A; Fig. 5a and Fig. S5B) while the levels of Tfr cells were similar (Fig. S5C). Furthermore, the deletion of *Camk4* in T cells led to a change in splenic B cell populations, with a decrease in the pathogenic age-associated B cells (B220⁺CD93⁻CD23⁻CD21/35⁻; ABC) and normalization of follicular B cells (B220⁺CD93⁻CD23⁺CD21/35⁺; FOB; Fig. 5b; Fig. S5D, E). Similar to our findings in the immunization model, GCB cells were significantly decreased in the spleen and cLNs from B6.*lpr.Camk4*^fl/fl^.*dlck-Cre* mice (Fig. 5c and Fig. S5F). Importantly, antibody-producing CD138⁺ plasma cells were also decreased in T cell-specific *Camk4*-deficient mice (Fig. 5d and Fig. S5G). These differences resulted in decreased total and subclass-specific anti-dsDNA antibodies (Fig. 5e, f). Hypergammaglobulinemia, a hallmark of B cell hyperactivation in autoimmunity, was

absent in B6.*lpr.Camk4*^fl/fl^.*dlck-Cre* mice (Fig. 5g). At the tissue level, we had previously shown that the B6.*lpr.Camk4*^fl/fl^.*dlck-Cre* were characterized by improved kidney pathology[8]. These mice also showed a significant decrease in the glomerular IgG and C3 deposition as evaluated by immunofluorescence (Fig. 5h, i). Overall, the deletion of the *Camk4* gene in T cells leads to decreased T_fh cells, normalization of B cell subsets, decreased auto-antibody production, and organ damage in systemic autoimmunity.

## CaMK4 controls BCL6 expression and T_fh cell function in healthy individuals and patients with SLE

To provide translational value of our findings from mice, we next sorted T_fh (CD4⁺CXCR5⁺CD25⁻CD127⁺), B memory (CD19⁺IgD⁻CD27⁺), and B naïve (CD19⁺IgD⁺CD27⁻) cells from healthy donors using FACS (Fig. S2A, B). Using reverse transcriptase qPCR, we found that human T_fh cells expressed significantly more *CAMK4* compared to B cells (Fig. 6a), similar to our findings in mice. Primary human T_fh cells cultured for 48 h with a CaMK4 inhibitor (KN93) significantly downregulated the expression of *BCL6* and *IL21* ($p < 0.05$, Fig. 6b), while T_fh activation (as assessed by *PDCD1* expression) was not affected (Fig. 6b). We confirmed at the protein level that IL21 production was significantly downregulated upon CaMK4 inhibition (Fig. 6c). To confirm that the impact on BCL6 expression was mediated by CAMK4 inhibition, we transfected human primary Tfh cells with CAMK4 target siRNA which led to BCL6 downregulation (Fig. S6). To investigate

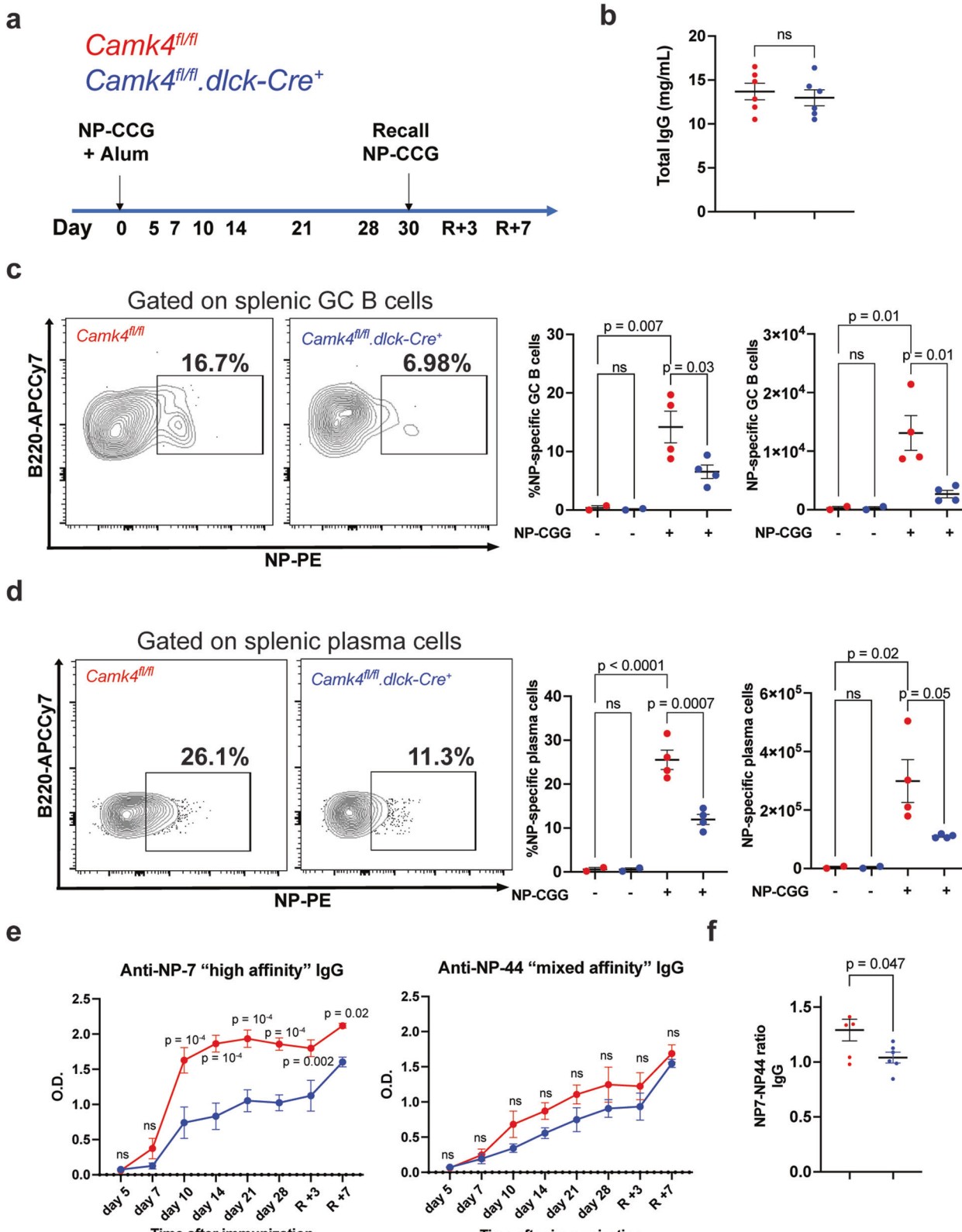

**Fig. 4 | T cell CaMK4 drives antigen-specific IgG response to immunization.** *Camk4^fl/fl* (red, *n* = 4) and *Camk4^fl/fl.dlck-Cre* (blue, *n* = 4) mice were immunized with NP-CGG and bled at different time points to evaluate antibody response. Non-immunized *Camk4^fl/fl* and *Camk4^fl/fl.dlck-Cre* were used as control (*n* = 2 per group). On day 30, a recall immunization was conducted. **a** Immunization and bleeding protocol. **b** Total serum IgG was measured at the end of the experiment using ELISA. **c**, **d** Gating strategy for NP-specific GC B cells (**c**) and NP-specific plasma cells (**d**) in immunized mice (left panel), and cumulative relative (middle panel) and absolute (right panel) number of cells. **e** Optical density of anti-NP-7 "high affinity" (left panel) and anti-NP-44 "mixed affinity" IgG (right panel) at different time points after immunization was assessed using ELISA. **f** The NP7/NP44 ratio is given for each immunized mouse. Unpaired two-tailed Student's *t*-test (**b**, **f**), paired one-way ANOVA with Holm−Sidak's correction (**c**, **d**) or two-way ANOVA with Holm−Sidak's correction (**e**). Bars indicate mean ± s.e.m.

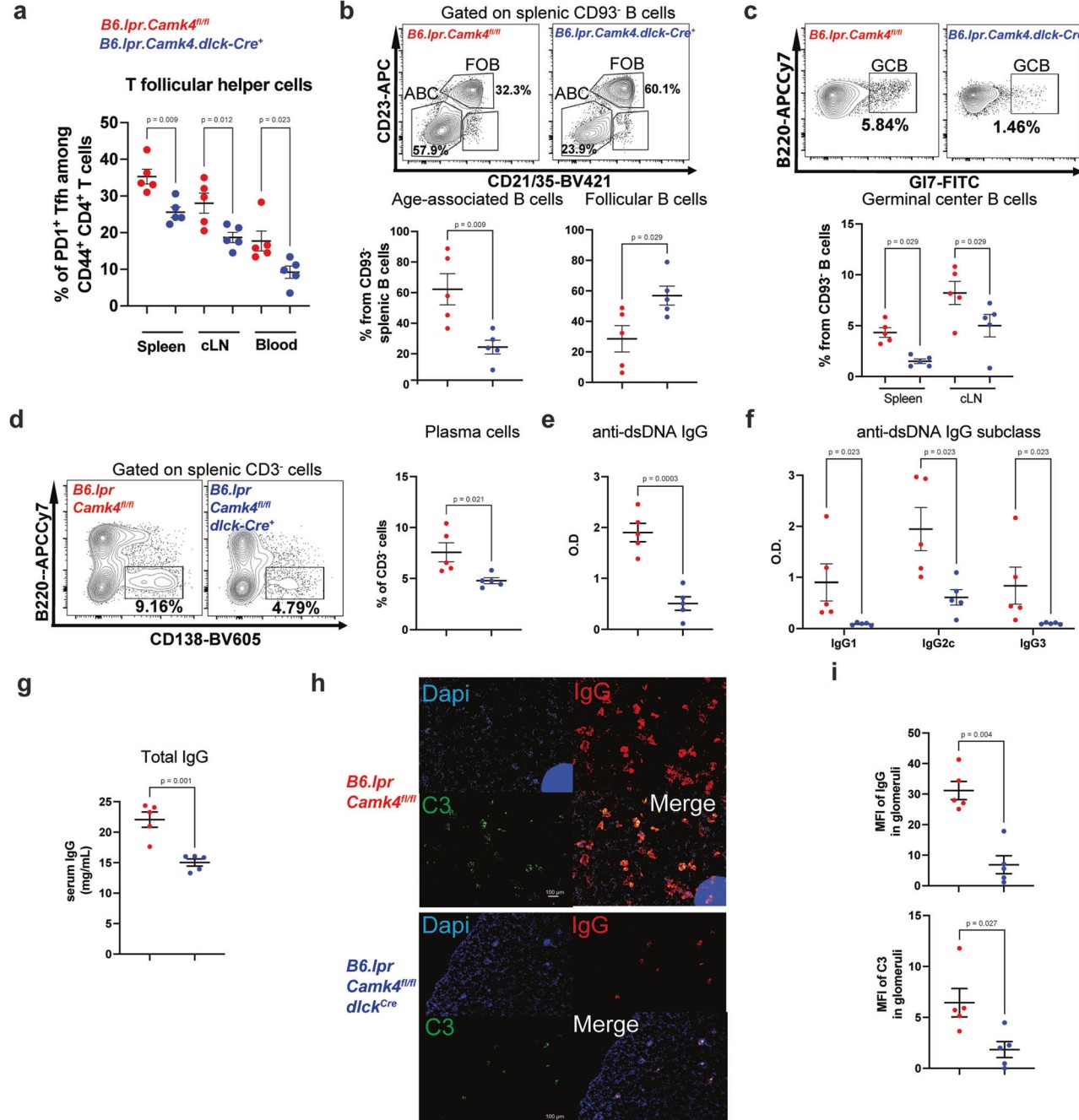

**Fig. 5 | Deletion of CaMK4 in T cells corrects B cell dysregulation and improves autoimmunity.** A 34-week-old B6.*lpr.Camk4^fl/fl* (red, *n* = 5) and B6.*lpr.Camk4^fl/fl*.*dlck-Cre* (blue, *n* = 5) were sacrificed, and the spleen, cervical lymph nodes (cLN), and blood were evaluated. **a** Percentage of PD1⁺ T_fh cells in the spleen, cLN, and blood. **b** Gating of splenic B cell subpopulations (upper panels) and cumulative results (lower panels). **c** Gating of splenic germinal center B (GCB) cells (upper panels) and cumulative data (lower panel). **d** Gating of splenic CD138⁺ plasma cells (left panels) and cumulative results (right panel). **e** Total anti-double stranded (ds) DNA IgG was measured using ELISA. **f** Isotype-specific anti-dsDNA IgG was measured using ELISA. **g** Total serum IgG level was measured using ELISA. **h** Kidney tissue was stained for IgG (PE, red), C3 (FITC, green), and DAPI (blue). Representative staining. White bars indicate 100 μm. **i** For each mouse, 20 glomeruli were identified, and the mean fluorescence intensity of IgG and C3 was measured. One-way ANOVA with Holm-Sidak's correction (**a**, **c**, **f**) or unpaired two-tailed Student's *t*-test (**b**, **d**, **e**, **g**, **i**). Bars indicate mean ± s.e.m.

whether the T_fh cell function was affected by CaMK4 inhibition, we cocultured freshly sorted human T_fh cells and B memory cells together with staphylococcus enterotoxin B (SEB). After 7 days of coculture, the percentage of CD38⁺CD27⁺ plasmablasts was significantly decreased in the presence of the CaMK4 inhibitor KN93 (Fig. 6d). Furthermore, the IgG production as measured in the coculture supernatant was significantly decreased in the presence of KN93 (Fig. 6e). Finally, we sought to evaluate the relevance of these findings in human SLE. We

sorted T_fh, Treg, and B cells from frozen PBMCs of 13 patients with SLE (Table S1). *CAMK4* expression was significantly higher in the T_fh cells compared with the other subsets (Fig. 7a). There was no difference in Tfh cells *CAMK4* expression between healthy donors and SLE patients (Fig. S7A). As previously shown[20], CAMK4 expression is not affected by SLE disease activity (Fig. S7B), whereas its activation and nuclear localization (which mediates its genetic effect) is[20]. In SLE T_fh cells, there was a significant correlation between *CAMK4* and *BCL6*

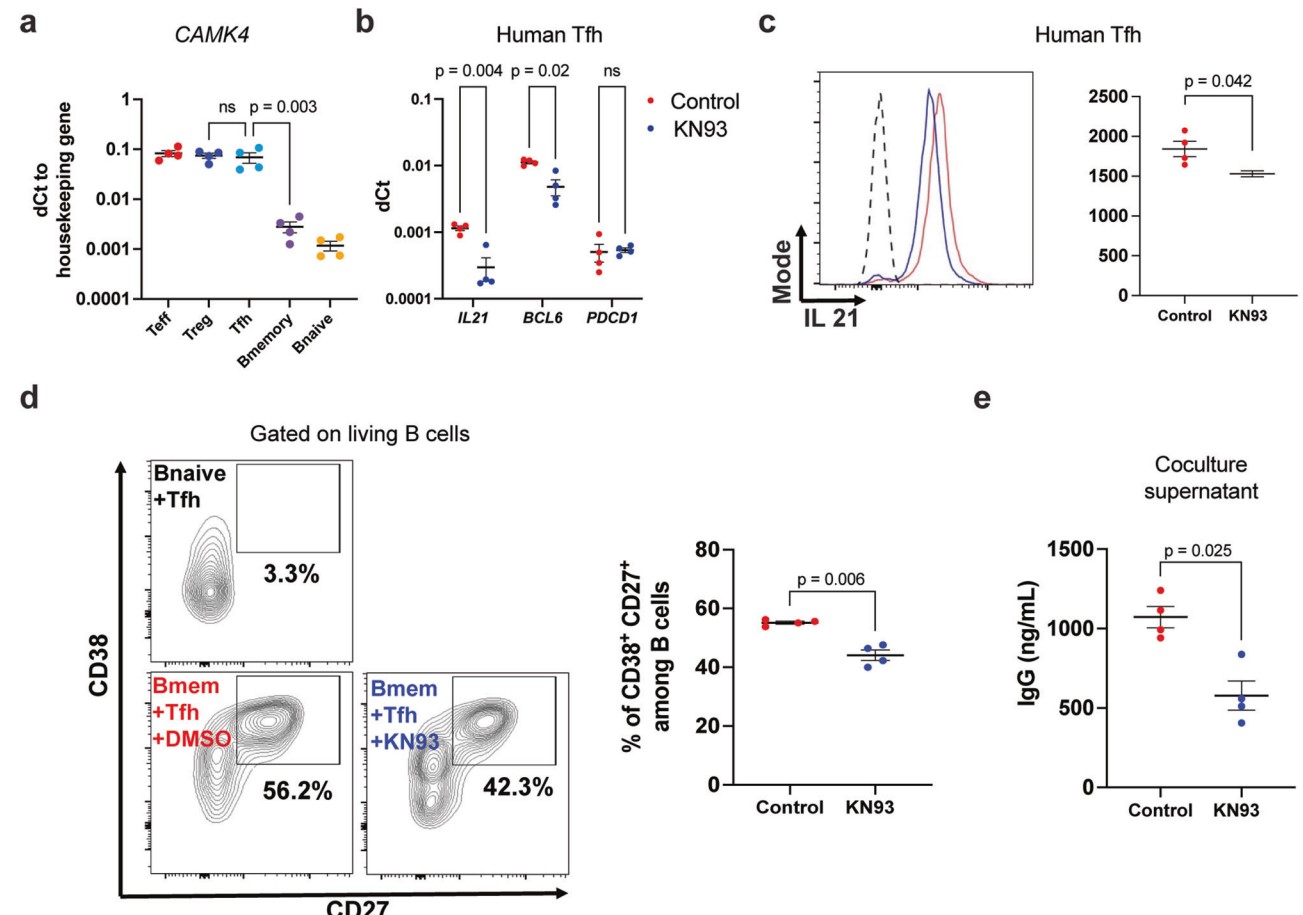

**Fig. 6 | CaMK4 modulates human T$_{fh}$ cell function. a** T follicular helper (T$_{fh}$, CD3$^+$CD4$^+$CD25$^-$CD127$^-$CXCR5$^+$) cells, T regulatory (T$_{reg}$, CD3$^+$CD4$^+$CD25$^+$CD127$^-$CXCR5$^-$) cells, T conventional (T$_{conv}$, CD3$^+$CD4$^+$CD25$^-$CD127$^+$CXCR5$^-$) cells, B memory (CD3$^-$CD19$^+$CD27$^+$IgD$^-$) and B naïve (CD3$^-$CD19$^+$CD27$^-$IgD$^+$) cells were sorted from healthy donors peripheral blood mononuclear cells ($n = 4$ donors). RNA was isolated, and reverse-transcriptase qPCR was conducted for *CAMK4* expression using GAPDH as a housekeeping gene. **b, c** Freshly sorted healthy donor's T$_{fh}$ were cultured for 48 h with CD3/CD28 stimulation and a CaMK4 inhibitor KN93 (10 μM) or control (DMSO). **b** RNA was isolated, and reverse-transcriptase qPCR conducted for *IL21*, *BCL6*, and *PDCD1* expression. **c** T$_{fh}$ cells

were stimulated with PMA/ionomycin/GolgiStop and stained for IL21. Representative staining (left panel) and cumulative results (right panel, $n = 4$ mice). **d** Freshly sorted healthy donor T$_{fh}$ were cocultured with autologous B naïve or B memory cells (1:1) together with staphylococcal enterotoxin B (SEB) ± KN93 (10 μM) or DMSO for 7 days. After coculture, representative gating of the differentiation of activated plasmablast (CD27$^+$CD38$^+$) B cells (left panel) and cumulative results (right panel, $n = 4$ independent experiments). **e** Human IgG was measured in the coculture supernatants using ELISA ($n = 4$ independent experiments). One-way ANOVA with Holm−Sidak's correction (**a**, **b**) or paired two-tailed Student's *t*-test (**c**–**e**). Bars indicate mean ± s.e.m.

expression ($r^2 = 0.626$, $p = 0.002$; Fig. 7b), suggesting that CaMK4 may control *BCL6* expression in human SLE in the same way that we described in mice.

## Discussion

In the current study, we have demonstrated that T cell-specific CaMK4 expression drives T$_{fh}$ cell expansion and, ultimately, humoral immune response during immunization and nominal antigens and autoimmunity. A T cell-specific deletion of *Camk4* led to impaired T$_{fh}$ cell expansion, decreased germinal center formation, and overall antibody responses after immunization with T cell-dependent antigens. Mechanistically, we found that CaMK4 controls the transcription of the *Bcl6* lineage T$_{fh}$-determining transcription factor through the activation of CREMα. Using human primary T$_{fh}$ cells, we confirmed that CaMK4 was necessary for the B cell helper function of T$_{fh}$.

CaMK4 is expressed in multiple subsets of CD4$^+$ T cells, and its activation is also involved with the differentiation and function of Th17[7,21] and Treg cells[8,9]. In the immunization model, we used a mouse with a genetic deletion of *Camk4* in all T cells, which opens the possibility that other subsets of T cells were affected by this deletion.

However, we did not find any changes in the levels of T$_{reg}$ nor T$_{fr}$ cells (Fig. 1d), supporting the fact that the differences observed in terms of B cell responses are linked to the role of CaMK4 on the T$_{fh}$ cell compartment.

Interestingly, our results are different from those observed by Nakaya and coauthors[22]. They observed that the day 3 expression of *CAMK4* in PBMCs from individuals immunized with the inactivated influenza vaccine negatively correlated with the vaccinal antibody response at day 28[22]. However, a recent transcriptomic atlas encompassing more than 3000 individuals immunized with 13 different vaccines failed to identify an association between *CAMK4* expression (or any other gene) and antibody response[23]. Furthermore, Haralambevia and colleagues found that T-cell *CAMK4* expression was (non-significantly) negatively associated with the Fluad vaccine antibody response and (non-significantly) positively associated with the Fluzone vaccine antibody response[24]. Nakaya and colleagues[22] also showed that mice with a *Camk4* germline deletion developed enhanced IgG response to flu vaccines, which is in contrast to our results with NP-CGG immunization (Fig. S4D). The differences might be explained by the different routes of immunization (intraperitoneal

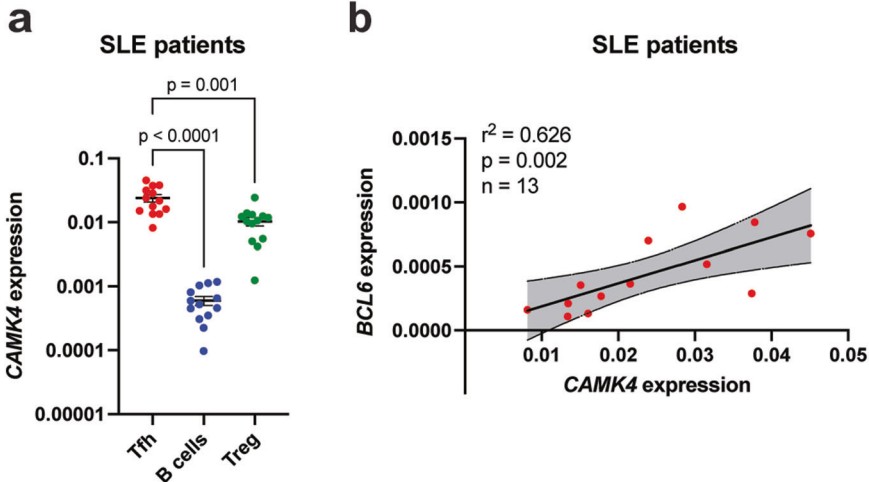

**Fig. 7 | CaMK4 expression in SLE $T_{fh}$ cells is increased compared with $T_{reg}$ cells and correlates with $T_{fh}$ *BCL6* expression.** T follicular helper (CD19$^-$CD4$^+$CD25$^-$ CD127$^+$CXCR5$^+$; $T_{fh}$) cells, T regulatory (CD19$^-$CD4$^+$CD25$^+$CD127$^-$Treg) cells, and B (CD4$^-$CD19$^+$) cells were sorted from frozen PBMCs of SLE patients ($n = 13$, Table S1). **a** RNA was isolated, and reverse-transcriptase qPCR was conducted for *CAMK4* expression. **b** Spearman correlation between *CAMK4* and *BCL6* expression in SLE $T_{fh}$ cells. The shaded area indicates the 95% confidence interval of the correlation. Paired one-way ANOVA with Holm–Sidak's correction for multiple analysis. Bars indicate mean ± s.e.m.

vs. intramuscular injections)[25] or the presence of adjuvant (Alum in our study *vs.* no adjuvant). Indeed, different vaccination routes result in different diffusion of the antigen to secondary lymphoid organs that might impact the T/B cell response to the antigen.

CREMα belongs to a superfamily of transcription factors, which also includes the inducible cAMP early repressor (ICER) and the cAMP-responsive element binding proteins−1 and 2 (CREB-1/2)[26]. All these transcription factors recognize the CRE-binding site and result in either the upregulation or the downregulation of the gene. For example, CaMK4 activation of CREMα is responsible for the upregulation of the *Il17a* gene, which promotes the expansion of the Th$_{17}$ subset in autoimmunity[7,27]. Conversely, CaMK4 activation of CREMα leads to the downregulation of the *Il2* gene, which results in the impairment of Treg cell expansion and function and may worsen autoimmunity[9,20]. Importantly, *CAMK4* gene expression is similar between healthy donors and SLE patients (Fig. S6A), but CaMK4 protein phosphorylation (i.e., activation) and nuclear localization are increased in SLE[8,20]. In the current study, we found that the presence of the CRE-binding site is necessary for the transcription control of *Bcl6* in murine $T_{fh}$ cells and confirmed using *Icer/Crema*$^{-/-}$ mice that this effect is mediated by ICER/CREMα and not through CREB.

Although SLE is viewed as an auto-antibody-mediated disease, strategies aiming at the containment of the B cells compartment, notably with rituximab or other B cell-depleting antibodies, have been disappointing[28,29]. Advances in our understanding of the molecular mechanisms underlying T cells and B cell dysregulation in autoimmunity have led to the development of new targeted treatments. These include targeted synthetic disease-modifying anti-rheumatic drugs (tsDMARD) such as Janus kinase (JAK), non-receptor tyrosine kinase 2 (Tyk2) or Bruton's tyrosine kinase (BTK) inhibitors[4]. This study, along with previous studies from our team, suggests that the inhibition of CaMK4 signaling represents a valuable strategy in the treatment of patients with SLE. Indeed, CaMK4 is involved in the dysregulation of several subsets of T cells, including $T_{fh}$, $T_{reg}$[8,9], and Th$_{17}$ cells[7,21], but also in the promotion of kidney damage through its expression in podocytes[30,31]. The main issue with tsDMARD are the off-target effects. For example, these off-target effects may be responsible for elevated risk of thrombosis and cancer of the JAK inhibitor Tofacitinib[32]. CaMK4 is also expressed in the hippocampus[33], underlying the importance of developing compounds or drug delivery

strategies that do not cross the blood-brain barrier, such as CD4-tagged nanolipogels loaded with a CaMK4 inhibitor[34].

The strengths of our study include the confirmation of our findings in mice using primary human $T_{fh}$ cells and the study of a group of SLE patients, with nearly half of them having high disease activity at the time of the sampling (Table S1). The main limitation of our study is the use of mice with deletion of *Camk4* in all T cells and not only in $T_{fh}$ cells. For example, restored IL-2 production in CaMK4-deficient mice may participate in Tfh cell dysfunction and impaired follicular center formation[35]. However, we believe that CaMK4 activation is an early event during naïve T cell activation and differentiation. Therefore, the deletion of CaMK4 in fully differentiated T cells by using *Camk4*$^{fl/fl}$.*Bcl6-Cre* mice may not recapitulate the events that happen during the generation of $T_{fh}$ cells while a normal or autoimmune response evolves.

In conclusion, this study identifies CaMK4 activation in T cells as a major factor responsible for the expansion of T follicular helper cells and a full-blown T-dependent humoral response in the setting of immunization and autoimmunity. These findings reinforce the rationale for CaMK4 inhibition as a multitarget therapy in autoimmune diseases such as SLE.

## Methods
### Human samples
Healthy donors and patients with a diagnosis of SLE fulfilling the ACR/EULAR 2019 classification criteria of SLE were recruited in the study. All participating volunteers provided written informed consent. The Beth Israel Deaconess Medical Center Institutional Review Board approved this study protocol. Peripheral blood was drawn in heparin-lithium tubes and processed either fresh or after thawing of previously frozen PBMCs. Blood collars from healthy blood donors were obtained from the Boston Children's Hospital donor center.

### Mice
The animals were housed in a pathogen-free Animal Facility in the Beth Israel Deaconess Medical Center (BIDMC, Center for Life Science). B6.129 × 1-Camk4tm1Tch/J (*Camk4*$^{-/-}$, stock #004994) and B6.MRL-Faslpr/J (B6.lpr; stock #000482) and C57Bl/6 (stock #000664) were obtained from the Jackson laboratory. B6.*Camk4*$^{fl/fl}$.*dlck*$^{Cre}$ mice obtained from previous studies[8], and were crossed with the B6.MRL-

Faslpr/J to obtain B6.lpr.*Camk4*[fl/fl].*dlck*[Cre]. The SV129/Bl6.*Icer/Crema*[−/−] mice were obtained from Günther Schuetz (Das Deutsche Krebsforschungszentrum, Heidelberg, Germany)[36] and were backcrossed with C57BL/6 mice for over nine generations. Immunization of 8-week-old mice of both sexes was conducted by the i.p. injection with either 100 μL of sheep red blood cells (Thermofisher, Waltham, MA, USA), or 25 μg of Alum-conjugated 4-Hydroxy-3-nitrophenylacetyl chicken γ-globulin (NP-CGG, ratio 20-29; Biosearch technologies, Petaluma, CA, USA) diluted in sterile PBS. The recall immunization was conducted with an i.p. injection of 50 μg of NP-CGG diluted in sterile PBS. Experimental and control animals were bred together (B6.*Camk4*[fl/fl].*dlck*[Cre] and B6.lpr.*Camk4*[fl/fl].*dlck*[Cre]) or in separate cages (other strains). Euthanasia was conducted using $CO_2$ inhalation following the recommendations from the Institutional Animal Care and Use Committee. The animal study protocol was approved by the BIDMC Institutional Animal Care and Use Committee.

## Cell culture and in vitro T cell differentiation

Murine CD62L[+]CD4[+] T cells were isolated using magnetic cell sorting from mouse splenocytes according to the manufacturer's instruction (Miltenyi Biotec, Charlestown, MA, USA). In total, $3 \times 10^5$ cells were cultured in a 48-well plate coated with goat anti-hamster crosslinking antibody in RPMI 1640 medium complemented with penicillin/streptomycin, 10% fetal bovine serum, and ß-mercaptoethanol. To differentiate the cells into T$_{fh}$, the medium was supplemented with anti-CD3 (0.25 μg/ml, 145-2C11; BioLegend, San Diego, CA), anti-CD28 (0.5 μg/ml, 37.51; BioXcell, Lebanon, NH), interleukin-21 (50 ng/mL, R&D systems, Minneapolis, MN, USA) interleukin-6 (100 ng/mL, R&D systems), anti-TGFß (0.5 μg/mL, clone 1D11; R&D systems) anti-IFN-gamma (10 μg/ml; BioXcell, Lebanon, NH, USA), and anti-IL-4 (10 μg/ml; BioXcell). The cells were retrieved for analysis on day 3 of the culture.

For the coculture experiment using human T$_{fh}$ and B cells, 50,000 freshly sorted autologous T$_{fh}$ and B cells were cocultured at a 1:1 ratio in a 96-well plate with staphylococcal enterotoxin B (SEB, 1 ng/mL, Fisher Scientific) for 7 days at +37 °C. The CaMK4 inhibitor KN93 (Selleckchem, Houston, TX, USA) or its vehicle (dimethysulfoxide, DSMO) was used at the concentration of 10 μM.

## Western blotting

Cell lysates were loaded on NuPAGE 10% Bis−Tris Gel (ThermoFisher Scientific) for electrophoresis and later transferred to a nitrocellulose membrane. The membrane was blocked with skimmed milk 6% for 1 h and incubated with the antibodies. The primary antibodies used were rabbit polyclonal anti-Bcl6 (ThermoFisher Scientific, Ref PA5-27390, dilution 1:1000) and mouse anti-ß-actin (Sigma-Aldrich, clone AC-74, dilution 1:2500). The secondary antibodies were goat anti-rabbit horseradish peroxidase- (HRP-) conjugated antibody (ThermoFisher Scientific, reference 31460) and an anti-mouse IgG HRP-conjugated antibody (Abcam, reference ab6789). The membranes were revealed using an ECL system (Cytiva, Marlborough, MA) using a ChemiDoc XRS + (Bio-Rad, Hercules, CA USA), and the obtained picture was analyzed using the ImageJ software (NIH).

## Flow cytometry and cell sorting

Surface staining was conducted in phosphate-buffered saline (PBS) at room temperature for 15 min except for CXCR5 staining, which was incubated for 60 min at +4 °C. For intracellular staining, the cells were fixed with Cytofix/Cytoperm buffer (BD Bioscience, San Jose, CA, USA) according to the manufacturer's instruction and stained overnight in Perm/wash buffer at +4 °C. For cytokine staining, the cells were cultured for 4 h in fully supplemented RMPI along with phorbol myristate acetate (500 ng/mL; Sigma-Aldrich), ionomycin (1.4 μg/mL; Sigma-Aldrich, Saint-Louis, MO, USA) and GolgiStop (BD Biosciences). A list of the antibodies used in this study is found in Table S2. The acquisition was conducted using Cytoflex (Beckman Coulter, Brea, CA, USA).

Fluorescence-activated cell sorting (FACS) was conducted using a FACSAria (BD Biosciences). For human cell FACS, CD4[+] T cells and total B cells were presorted from a blood collar using RosetteSep human CD4[+] T cells or human B cells (Stemcell technologies, Kent, WA, USA), respectively, according to the manufacturer's protocol. Initial gating strategies are shown in Fig. S8.

## ELISA

Specific IgG was measured using an in-house enzyme-linked immunosorbent assay (ELISA)[37]. Briefly, the target protein was coated on a MaxiSorp Nunc ELISA plate (Sigma-Aldrich) overnight (NP-7 and NP-44: 20 μg/mL; Calf DNA 50 μg/mL; Sigma-Aldrich). In the case of calf DNA, the plates were pre-coated with poly-L-lysin (50 μg/mL) for 2 h at room temperature. After blocking with PBS-BSA 1% for 2 h, the serum diluted in PBS−BSA 1% was incubated for 2 h at room temperature with shaking. The plate-bound IgG were revealed using an alkaline phosphatase-conjugated anti-mouse IgG (Jackson Immuno Research; reference 1:5000) or with an alkaline phosphatase-conjugated anti-mouse IgG1, 2c, or 3 (Jackson ImmunoResearch, West Grove, PA, USA; references 115-055-205, 115-055-208, and 115-055-209, respectively) for 1 h. The colorimetric reaction was started using diethanolamine substrate buffer (ThermoFisher Scientific) and p-Nitrophenyl phosphate phosphatase substrate, and the optical density was read at 405 nm.

## Site-directed mutagenesis, vector, and siRNA transfection

A construct with the *Bcl6* gene promoter the luciferase reporter (pGL3_bcl6_vector), and the CaMK4 overexpression vector was obtained from Genscript. The list of vectors used in the study is given in Table S3. Site-directed mutagenesis was conducted to induce deletion of the −62/−58 CRE binding site (CGTCA) using the Q5 site-directed mutagenesis kit (New England Biolabs, Ipswich, MA, USA) with the following primers: forward, 5′- GGAGCCCACGTGACGGCG-3′ and reverse, 5′-TCGGCGTCTTCGCTGTAGC-3′. The deletion was confirmed with Sanger sequencing (Azenta) using the RVprimer 3 primer. The cells were transfected at day 2 of differentiation with 10 μg of the vector using an Amaxa Nucleofector device (Lonza, Basel, Switzerland). The program X-001 was used with the Amaxa Mouse T cell nucleofector kit (Lonza) according to the manufacturer's protocol. The results were assessed at day 3 (24 h after transfection).

For siRNA transfection, primary cells were transfected with 20 nM of SMARTpool CAMK4 siRNA (Horizon discovery®) using an Amaxa Nucleofector device (list of siRNA in Table S2). The program T-023 was used with the Amaxa Human T cell nucleofector kit (Lonza) according to the manufacturer's protocol. After transfection, the cells were rested for 6 h in total RPMI and subsequently cultured on 96-well plates coated with anti-CD3 (2 μg/mL) in complete RPMI supplemented with soluble anti-CD28 (2 μg/mL) for 42 h before RNA extraction.

## Luciferase assay

The firefly luciferase reporter vector with the wild-type or mutated Bcl6 promoter was transfected (10 μg of vector per transfection) along with 200 ng of a Renilla luciferase vector used as internal control. The transfection was conducted on day 2, and the luciferase activity was assessed on day 3 using the Promega Dual Luciferase Assay System (Promega, Madison, WI, USA) according to the manufacturer's instructions.

## Chromatin immunoprecipitation (CHIP-) PCR

Inducible Tfh cells were differentiated from the CD62L[+]CD4[+] T cells from C56Bl/6 or *Camk4*[−/−] mice, as described previously. Cells were lysed, and immunoprecipitation was conducted using the MAGnify Chromatin Immunoprecipitation System (Thermo Fisher Scientific) with anti-CREMα (Santa-Cruz; Table S2) or control antibody according to the manufacturer's instructions. qPCR was conducted using a

primer specific for the *Bcl6* gene promoter region (Table S3). Relative binding was calculated by the comparison ($\Delta$Ct) between CREM$\alpha$ and control IgG with the formula $2^{-deltaCt}$.

## Immunofluorescence

Mouse tissues, including both kidneys and spleens, were embedded in optimal cutting temperature (OCT) tissue media (Tissue-Tek, Torrance, CA, USA) and frozen on dry ice. Coronal 6 $\mu$m sections were cut using a cryotome and blocked using PBS-BSA 5%. Mouse kidneys were stained with anti-mouse IgG phycoerythrin (T-6390, Invitrogen, Carlsbad, CA, USA; dilution 1:100), fluorescein-conjugated anti-C3 (SKU 0855500, MP Biomedicals, Irvine, CA, USA; dilution 1:100) and mounted with ProLong Gold antifade mountant with DAPI (Thermofisher). The image acquisition was conducted using a BZ-X800 analyzer. The mean fluorescence intensity of each marker was assessed for 20 glomeruli per mouse. Spleens were stained with Biotin-conjugated anti peanut allergen (anti-PNA, Vector lab, reference 1075-5), Alexa Fluor 647 anti-mouse/human B220 (Biolegend, reference 103226), Alexa Fluor 488 anti-mouse CD4 (Biolegend, reference 100529). Secondary staining was done using Streptavidin-Alexa Fluor 405 conjugate (ThermoFisher, reference S32355). The slides were mounted with ProLong Gold antifade mountant (Thermofisher), and the image acquisition was conducted using a Zeiss LSM 780 confocal microscope.

## Quantitative PCR

The RNA was isolated with the TRIzol reagent (Sigma-Aldrich), and complementary DNA was retrotranscribed from RNA using cDNA EcoDry tubes (Takara, Mountain View, CA, USA) following the manufacturer's instructions. The qPCR was conducted using Taqman probes (see list in Table S3), Taqman reaction buffer, and cDNA according to the manufacturer's instruction and evaluated with a LightCycler 480 thermocycler (Roche, Indianapolis, IN, USA).

## Statistical analysis

The figures and statistical analysis were made using GraphPad Prism Version 9.4.0. In the figures, the points indicate an individual data point (independent experiment or mouse), and bars represent mean ± SEM. Quantitative data were compared using Student's two-tailed *t*-test for 2 groups or one-way analysis of variance (ANOVA) together with Holms−Sidak's correction unless stated otherwise. The non-parametric Spearman correlation test was used for correlation studies. To analyze the IgG response over time, we used a two-way ANOVA, taking into account sampling time and the mice (matched samples). Multiple testing was corrected using the Hold−Sidak's correction. A *p*-value < 0.05 was considered statistically significant.

## Reporting summary

Further information on research design is available in the Nature Portfolio Reporting Summary linked to this article.

## Data availability

Source data are provided in this paper. Uncut Western blot membranes are provided in the source data file. The raw data that support the findings of this study are available from the corresponding author upon request. Source data are provided in this paper.

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

## Acknowledgements
This work was funded by National Institutes of Health grant R37 AI49954 (to G.C.T.). M.S. was financially supported by the Société Française de Rhumatologie, Arthur-Sachs & Monahan fellowships, and the Philippe Foundation. M.S. current position (France) is supported by the Institut National pour la Santé et la Recherche Médicale (INSERM), the Bettencourt-Schueller Foundation, the ATIP-Avenir program and the Foundation for Research in Rheumatology (Foreum).

## Author contributions
M.S., H.L. and G.C.T. conceptualized and designed the study. M.S., H.L., W.P., N.Y., K.K., W.L. T.V., and M.G.T. performed the experiments, data analysis, and interpretation. G.C.T. and A.B. provided the human samples. M.S. and G.C.T. prepared the paper.

## Competing interests
The authors declare no competing interests.
