## [Peer Review File · Nature Communications]

REVIEWER COMMENTS

Reviewer #1 (expert in genomics of SLE and therapeutic targets for SLE):

The study by Scherlinger et al. examined the regulatory role of CaMK4 (calcium/calmodulin-dependent protein kinase IV) in T-cell-dependent humoral response. The authors demonstrated that CaMK4 was crucial for Tfh cell differentiation by regulating Bcl6 gene expression through CREM α . The resulting decrease in Tfh cells due to CaMK4 deficiency led to reduced germinal center formation and antibody production. The correlation analysis of CaMK4 and BCL6 expression in SLE Tfh cells suggested that CaMK4 drives Tfh expansion in both humans and mice. Inhibition of CaMK4 caused human Tfh cell dysfunction in vitro, providing novel and significant insights into the etiology of lupus. The study was well-conducted and comprehensive, utilizing both mouse and human Tfh cells.

Major concerns :

1. The title "CaMK4 controls normal and autoimmune T-dependent B cell responses" of this study is misleading as the regulatory role of CaMK4 was T-cell intrinsic. Similarly, many of results and conclusions that Camk4 controls B cells development and humoral responses should be stated more clearly as T-cell intrinsic to avoid this misunderstandings that CaMK4 directly regulates B cells.
2. To further confirm that CaMK4 could bind with CREM α and modulate Bcl6 gene expression, CHIP-qPCR verification for these target genes in primary Tfh cells should be added.
3. CaMK4 deficiency/inhibition led to the downregulation of Bcl6 expression, thus regulating Tfh expansion during T-dependent humoral response. Whether CaMK4 overexpression is sufficient for Bcl6 expression and Tfh development? The authors may consider gain-of-function experiment by overexpressing CaMK4 in Tfh cells to test the effects on Bcl6 expression.
4. In figure 3B, instead of displaying the image alone, the quantity analysis of the GCs (eg the numbers/areas of GCs) should be conducted. Besides, the IgG subclass in NP-CGG induced model should be detected.
5. Detection of the percentage of NP-specific B cells (especially the NP-specific GCBs) in Camk4fl/flDclck-Cre and Camk4fl/fl mice should be added.
6. Excepted for the impaired antibody response, whether the affinity maturation was affected in Camk4fl/flDclck-Cre mice. The authors should compare the ratio of high affinity versus low affinity anti-NP antibody.
7. Whether the expression level of CaMK4 in SLE Tfh cells is correlated with disease activity (eg anti-dsDNA antibody)? Could the authors compare the expression of CaMK4 between healthy controls and SLE patients?

Minor concerns:

1. In figure 2E, luciferase reporter vector was used to investigate the molecular mechanism of CaMK4 in Tfh cells. The authors should show the transfection efficiency. The details of nucleofection in primary T cells should be completed in the methods.
2. Numbers given for the amount of mice used per group or replicates of experiments should be added in Figure 1, 2, 3 and 6.
3. In figure 4C-D, the authors used two-way ANOVA for statistical analysis, it should be described in the Materials and Methods part.
4. The gating strategy for all represented FACS plots should be added (eg. Figure 2A, 5A).
5. Some results were less correlated with the whole manuscript (Figure 1D). It would be better to move these results in the supplementary figures to increase the logic and coherence

Reviewer #2 (expert in TFH/B cell responses in health and disease, SLE):

This paper continues the investigations from the Tsokos group on the role of CAMK4 in T cell effector functions in relation to lupus pathogenesis, since CAMK4 is over expressed in lupus T cells.

The present study showed that Camk4 is highly expressed in murine and human Tfh cells, and that CAMK4 directly binds to the Bcl6 promoter. Camk4-deficiency in T cells reduced Tfh development and subsequently that of GC B cells and humoral responses in responses to immunizations as well as in the B6.lpr model of autoimmunity. They also showed that purified human Tfh treated with a CAMK4 inhibitor had a reduced ability to help memory B cells to differentiate into plasma cells. These results add Tfh cells as an effector T cells that is regulated by CAMK4. This implies that the therapeutic effect of CAMK4 inhibition is multipronged and targets all the major T cell subsets that have been implicated in lupus pathogenesis.

The study is straightforward, and the results support the author's conclusions.

A few issues should be addressed:

What was the effect of CAMK4 deletion in T cells in B6.Lpr mice on Treg and Tfr cells? Did it affect IFN γ and IL-17 production? Although the deletion in Tfh cells is most likely to be responsible for the B cell phenotypes, the decreases immune complex deposition in the kidneys could be attributed to other T cell subsets.

The authors should discuss how the increased IL-2 production that they have previously reported following CAMK4 inhibition could also contribute to Tfh cell inhibition.

CAMK4 expression is very low in B cells. Why was CAMK4 expression in human Tfh cells compared to that in B cells rather than other T cell subsets in both healthy subjects? The levels of CAMK4 expression in Tfh cells should be compared between HS and SLE patients.

Minor concerns:

A common definition of ABCs is CD11c⁺ Tbet⁺

Reviewer #3 (expert in calmodulin-dependent kinases):

In this manuscript, Scherlinger et al. investigated the role of CaMK4 expressed in T cells on the development of T follicular helper cells (Thf) and antibody response. To this purpose, the authors leveraged T-cell specific CaMK4 KO mice (Camk4 fl/fl Dcl1-Cre) combined with Lupus-prone mouse strains (B6.lpr). Human blood samples from SLE and healthy individuals were also used in the study.

The authors demonstrated that CaMK4 expressed in T cells is required to drive development of Thf cells and sustain B cells expansion in response to antigenic stimulation. Mechanistically, they propose that CaMK4 expressed in T cells is required to activate CREM α and drive BCL6 upregulation in Thf. Deletion of CaMK4 in T cells was also found associated with improved clinical features in lupus-prone mice. Data from human blood samples with KN93 drug inhibitor suggested that CaMK4 controls the expansion/development of in vitro generated Thf. Based on these results, the authors propose CaMK4-CREM α as an important signal pathway controlling T cell-dependent B cell response in autoimmunity.

The results included in this manuscript are interesting and identify a new function of CaMK4 in Thf cell biology and B cell response in normal and lupus-prone setting. However, previous data of showing the function of CaMK4 on T cell development, SLE, and immune-mediated disorders limit the novelty of this manuscript. Specifically, the role of CaMK4 in T cell subsets development (e.g., Th17 and Treg) alongside the ability of this protein to regulate critical functions of immune and non-immune cell types (e.g., myeloid cells, epithelial cells, endothelial cells, and podocytes), which have relevant roles in the pathogenesis of SLE and other immune-mediated/autoimmune diseases have been widely reported (e.g., Koga et al. 2012; Koga et al. JCI 2014; Ferretti et al. 2018; Maeda et al. 2018; Bhargava et al. 2021; Scherlinger et al. 2022; Yong et al. 2022). Similarly, the ability of CaMK4 to modulate CREM α (and CREB) activity in T cells has been already previously documented (Koga et al. JCI 2014;). Lastly, the role of CaMK4 in the modulation of B cell response to vaccine has been previously investigated (Nakaya et al. 2012), while the discrepancies between the results from this earlier publication with data included in the current manuscript have not been sufficiently discussed/investigated. In conclusion, the finding that CaMK4 regulates Thf functions is interesting and novel, while at least in part, this study provides incremental pieces of evidence supporting the well-known role of CaMK4 in SLE/autoimmunity.

Major criticisms

1. The absolute cell count number should be provided and shown in all datasets/figures. This is an important finding to clarify/distinguish between direct and non-direct effects on cell subsets.
2. The complete immune profile of mice used in this study (basal and after antigenic stimulation) should also be provided.
3. KN93 is a nonspecific inhibitor of CaMK4 and this compound blocks additional relevant targets in T cells (e.g., CaMKII). The author should address the specificity of this compound in T cells by comparing the effects of KN93 treatment on lymphocytes isolated from WT and CaMK4-deficient mice.
4. A gene silencing approach should be used to assess the function of CaMK4 on human development of Thf cells.
5. Germline global deletion of CaMK4 has been associated with enhanced response to inactivated flu trivalent vaccine (Nakaya et al. Nat Immunol, 2012). On the opposite, data included in the present manuscript demonstrated that *Camk4* fl/fl *Dcl*-Cre mice secrete lower amount of antibody in response to SRBC and NP antigenic stimulation, compared to WT mice. Discrepancies between global and T cell-specific KO should be discussed or eventually addressed by comparing the effects of germline global CaMK4 KO and T cell specific deletion on antigenic stimulation, including flu trivalent inactivated vaccine.
6. The title should be revised to reflect more closely the results of the study. For example, it should be clearly stated that the study focusses on the role of CaMK4 expressed in T cells on development and antibody response in normal and SLE setting. A more general statement of the role of CaMK4 on autoimmunity would imply the assessment of global and cell-specific (e.g., T cell, myeloid cell) deletion of CaMK4 in normal/pathological antibody response, including additional autoimmune models.

Minors:

Figure 1.

- a) Panel C: CXCR5⁻ and CXCR5⁺ seem to express different levels of BCL6. However, the gating strategy does not take into consideration this possible relevant difference. Gate should focus on BCL6^{high} and results would include % and absolute cell number. Alternatively, BCL6 gene/protein expression in CXCR5⁻ and CXCR5⁺ cells should be assessed with additional methods (e.g. qPCR, immunoblot) using purified cells.

Figure 2:

1. Panel B: Please, clarify whether the expression of BCL-6 (MFI) refers to total CD4 T cells.
2. Panel C: Not cut gel should be provided, CAMK4 expression should be shown. Actin expression between WT and KO is significantly lower. Please, explain and eventually normalize BCL-6 expression using an additional/alternative housekeeping gene (e.g., Tubulin). Please, clarify if dots refer to technical or biological (from independent experiments) replicates.

Figure 3

1. Panel B: quantitative data on GC should be provided and shown in the figure.

Figure 5

1. Assessment of T cell subsets (Naïve, CM, EM, Treg, and Tfh), which include % and absolute cell number should be provided.
2. Panel H-I: green staining cannot be visible. Please, provide better images and clarify how Ig and C3 signals have been quantified.

Figure 6:

1. Panel A: specify which housekeeping gene has been used for normalization.
2. Panel B and D: Provide absolute cell count numbers.

Discussion, 337-344: This paragraph should be revised, including more recent evidence. Specifically, the authors mentioned that results from Nakaya study (Nakaya et al. 2012), showing a negative correlation between CaMK4 expression in PBMC at day 3 post-flu-vaccination and antibody response at day 28, may depend on B cell expansion and the corresponding dilution of T cells specific genes, which should include CaMK4. Although qualitatively correct, a more careful quantitative analysis of B cell expansion in vaccinated individuals (approximately 10^3 B cells per million of PBMC) would not provide a robust/reasonable support to this hypothesis. The authors also mentioned that a more recent extensive analysis, which analyzed the response against 13 vaccines in 3000 individuals (Hagan et al. 2022), does not confirm the original finding on CaMK4 downregulation in high-responder individuals. However, the authors should also mention that a major conclusion of that large study is the failure to identify a "universal" predictive signature between the different vaccines. On the other hand, CaMK4 expression was also not found positively associated with an enhanced B cell response, as predicted by the results of the present manuscript. Lastly, the negative correlation between CaMK4 expression and antibody titer has been recently documented in a recent study assessing the influenza A/H3N2 antibody response to high-dose influenza and adjuvanted influenza vaccine in older adults (Haralambieva et al. 2022). This study should be mentioned and discussed.

RESPONSE TO REVIEWERS' COMMENTS

Reviewer #1 (expert in genomics of SLE and therapeutic targets for SLE):

The study by Scherlinger et al. examined the regulatory role of CaMK4 (calcium/calmodulin-dependent protein kinase IV) in T-cell-dependent humoral response. The authors demonstrated that CaMK4 was crucial for Tfh cell differentiation by regulating Bcl6 gene expression through CREM α . The resulting decrease in Tfh cells due to CaMK4 deficiency led to reduced germinal center formation and antibody production. The correlation analysis of CaMK4 and BCL6 expression in SLE Tfh cells suggested that CaMK4 drives Tfh expansion in both humans and mice. Inhibition of CaMK4 caused human Tfh cell dysfunction *in vitro*, providing novel and significant insights into the etiology of lupus. The study was well-conducted and comprehensive, utilizing both mouse and human Tfh cells.

We thank the Reviewer for his positive assessment of our manuscript.

Major concerns :

1. The title "CaMK4 controls normal and autoimmune T-dependent B cell responses" of this study is misleading as the regulatory role of CaMK4 was T-cell intrinsic. Similarly, many of results and conclusions that Camk4 controls B cells development and humoral responses should be stated more clearly as T-cell intrinsic to avoid this misunderstandings that CaMK4 directly regulates B cells.

In accordance with the reviewer's point, which was also raised by Reviewer #3, we have modified the title as follows:

"CaMK4 controls Tfh expansion and function during normal and autoimmune T-dependent B cell response"

2. To further confirm that CaMK4 could bind with CREM α and modulate Bcl6 gene expression, CHIP-qPCR verification for these target genes in primary Tfh cells should be added.

To address the reviewer's question on the control of Bcl6 transcription via CaMK4/CREM α , we conducted CHIP-qPCR assay. Briefly, we differentiated *in vitro* CD62L⁺ CD4⁺ T cells isolated from the spleen of Wild-type or Camk4^{-/-} mice (n = 3 mice per group) to iTfh cells. After DNA isolation and fragmentation, we pulled down DNA with either a control or a CREM α -specific antibody and conducted qPCR for the Bcl6 gene promoter region (sequence provided in supplementary file). Relative binding was calculated by the comparison (DeltaCt) between CREM α and control IgG with the formula $2^{-\text{deltaCt}}$. Three technical replicates were conducted for all biological replicates. We found that CREM specifically binds to the Bcl6 promoter region, and that this binding is significantly decreased in Camk4^{-/-} mice. We have added these data in the revised manuscript (**Fig. 2F**).

Figure 2F: Chromatin immunoprecipitation (CHIP) was conducted on the DNA of murine primary Tfh cell using a CREMA or control antibody. The qPCR targeted the Bcl6 gene promoter region, and relative binding was calculated by comparing the deltaCt between CREMA or control antibody. Three technical replicates are shown for each biological replicates (n = 3 mice per group). ****, p < 0.0001, using one-way ANOVA with Holm-Sidak's correction.

3. CaMK4 deficiency/inhibition led to the downregulation of Bcl6 expression, thus regulating Tfh expansion during T-dependent humoral response. Whether CaMK4 overexpression is sufficient for Bcl6 expression and Tfh development? The authors may consider gain-of-function experiment by overexpressing CaMK4 in Tfh cells to test the effects on Bcl6 expression.

To address the reviewer's suggestion, we conducted murine Tfh *in vitro* differentiation of WT or Camk4^{-/-} T cells. We transfected the cells with either an empty or a CaMK4-overexpression vector. We studied Bcl6 expression in transfected (GFP⁺) cells. We found that CaMK4 overexpression leads to the upregulation of Bcl6 in both WT and Camk4^{-/-} Tfh cells. We have added this set of results in the revised manuscript (**Fig. S2B**, see below).

Fig S2B legend: CD62L⁺ CD4⁺ T cells from wild-type or Camk4^{-/-} mice (n = 4 mice per group) were differentiated to Tfh cells *in vitro* and transfected with an empty or CaMK4

overexpression vector. At day 3 of differentiation, *Bcl6* expression was evaluated in transfected (GFP⁺) CD4⁺ T cells. *, $p < 0.05$; **, $p < 0.01$ using one-way ANOVA with Holm-Sidak's correction.

4. In figure 3B, instead of displaying the image alone, the quantity analysis of the GCs (eg the numbers/areas of GCs) should be conducted. Besides, the IgG subclass in NP-CGG induced model should be detected.

To address the reviewer's comment, we have quantified the areas of GC in *Camk4^{fl/fl}* and *Camk4^{fl/fl}.dlck^{Cre}* mice (n = 3 per group). We have added these data in the revised manuscript (Fig. 3C-D).

Figure 3C-D legend: (C) Immunofluorescence of the spleen 7 days after immunization showing T cell (CD3, green), B cells (B220, red) and germinal centers (peanut antigen PNA, blue). (D) The area of germinal centers (PNA staining) was measured in the spleen of *Camk4^{fl/fl}* and *Camk4^{fl/fl}.dlckCre⁺* (n = 3 per group). Each dots represents the area of one germinal center. ***, $p < 0.001$ using Kolmogorov-Smirnov test.

Furthermore, we have conducted ELISA for NP-specific anti-NP7 and anti-NP44 IgG1, IgG2c and IgG3 (see below). We calculated the NP7/NPP4 ratio for IgG subtypes and show that the IgG1 and IgG3 subtypes were affected by T cell specific *Camk4* deletion. We have added these data in the revised manuscript (Fig. S4A-C).

A

Figure S4A-C: *Camk4^{fl/fl}* and *Camk4^{fl/fl}.dlck-Cre* mice ($n = 6$ per group) were immunized with NP-CGG and bled at different time points to evaluate the antibody response. ELISA results showing the optical density for the NP-7 "high affinity" IgG (left panel), NP-44 "mixed affinity" (middle panel) over time of *Camk4^{fl/fl}* (red) and *Camk4^{fl/fl}.dlck-Cre* (blue) mice. The right panels indicate the NP7-NP44 ratio 7 days after recall immunization.

(A) IgG1 results for anti-NP7 (left panel), anti-NP44 (middle panel) and the NP-7/NP44 ratio (right panel). (B) IgG2c results for anti-NP7 (left panel), anti-NP44 (middle panel) and the NP-7/NP44 ratio (right panel). (C) IgG3 results for anti-NP7 (left panel), anti-NP44 (middle panel) and the NP-7/NP44 ratio (right panel). Dots indicate the mean O.D. \pm s.e.m. *, $p < 0.05$; **, $p < 0.01$; ***, $p < 0.001$ using two-way ANOVA with Holm-Sidak's correction (longitudinal ELISA results) or Student's *t*-test (NP7-NP44 ratio).

5. Detection of the percentage of NP-specific B cells (especially the NP-specific GCBs) in *Camk4^{fl/fl}.dlck-Cre* and *Camk4^{fl/fl}* mice should be added.

In accordance with the reviewer's suggestion, we have immunized *Camk4^{fl/fl}* and *Camk4^{fl/fl}.dlck^{Cre}* mice ($n = 5$ per group) with NP-CGG following the same protocol as previously. After 14 days, we euthanized the mice, retrieved splenocytes and conducted cytometry to identify NP-specific splenic B cell subpopulations using NP-PE (Santa Cruz). We found that *Camk4^{fl/fl}.dlck^{Cre}* mice have decreased NP-specific B

cells subsets including GC B (B220⁺CD93-GI7⁺CD95⁺) and plasma (CD3⁻B220^{low}CD138⁺) cells. We have added these data in the revised manuscript (**Fig. 4C-D**).

C

D

Figure 4C-D: *Camk4^{fl/fl}* and *Camk4^{fl/fl} dlck-Cre⁺* mice were immunized with NP-CGG ($n = 5$ per group) and euthanized at day 14. NP-specific cells were studied among splenic B cells (gated as in Fig. 3A). *, $p < 0.05$; **, $p < 0.01$; ***, $p < 0.001$; ****, $p < 0.0001$ using one-way ANOVA with Holm-Sidak's correction.

6. Excepted for the impaired antibody response, whether the affinity maturation was affected in *Camk4^{fl/fl} dlck-Cre⁺* mice. The authors should compare the ratio of high affinity versus low affinity anti-NP antibody.

We have calculated the NP-7/NP-44 ratio for total IgG and IgG subtypes. We found that the ratio was decreased in mice with T-cell specific *Camk4* deletion for total IgG (**Fig. 4F**), IgG1 and IgG3 (**Fig. S4A-C**). These results further support the fact that B cell affinity maturation is affected in these mice, likely due to Tfh dysfunction. All these data have been added in the revised manuscript.

E

F

Figure 4E-F: (E) Optical density of anti-NP-7 "high affinity" (left panel) and anti-NP-44 "mixed affinity" IgG (right panel) at different timepoint after immunization assessed using ELISA. (F) The NP7/NP44 ratio is given for each immunized mice. *, $p < 0.05$; **, $p < 0.01$; ***, $p < 0.001$; ****, $p < 0.0001$ using unpaired Student's t-test (F) or two-way ANOVA with Holm-Sidak's correction (E).

A

B

C

Figure S4A-C: Optical density of anti-NP-7 "high affinity" (left panel) and anti-NP-44 "mixed affinity" IgG (middle panel) at different timepoint after immunization assessed using ELISA, NP7/NP44 (right panel). Results for IgG1 (A), IgG2c (B) and IgG3 (C).

7. Whether the expression level of CaMK4 in SLE Tfh cells is correlated with disease activity (eg anti-dsDNA antibody)? Could the authors compare the expression of CaMK4 between healthy controls and SLE patients?

We thank the reviewer for raising this important point. There is no correlation between CAMK4 gene expression and SLE disease activity index (see below, **Fig S7B** of the revised manuscript). Furthermore, the expression of CAMK4 is similar between healthy donors and SLE patients (**Fig. S7A**). These results are expected since we have previously shown that in patients with active SLE, the level of nuclear (active) CaMK4 protein compared to healthy control (Juang et al JCI 2005 & Scherlinger et al., Science Advances 2022).

Figure S7A-B: (A) Reverse transcriptase (RT-) qPCR of CAMK4 expression in human primary Tfh from healthy donor (n = 6) and SLE patients (n = 13). (B) Spearman's correlation between CAMK4 gene expression (assessed using RT-qPCR) and SLE disease activity index.

Figure 4 from Juang et al, JCI 2005: Increased expression of CaMKIV in the nucleus of SLE T cells. (A) Both cytoplasmic and nuclear proteins were isolated from normal and SLE T cells. Western blots were then conducted by sequential blotting of the membrane with antibody against CaMKIV or CaMKII. Representative blots from 2 patient-control pairs are shown. (B)

Eleven pairs were studied. Nuclear densitometric readings for CaMKIV, but not CaMKII, are higher ($P < 0.01$) in SLE T cells than in normal T cells. $*P < 0.05$

In accordance with the reviewer's suggestion, we have added the comparison between healthy and SLE Tfh expression of CAMK4, and the correlation between CAMK4 gene expression and SLE disease activity index (**Fig. S7A-B** of the revised manuscript). Furthermore, we have discussed this point in the revised manuscript.

Page 9, line 390-393: "There was no difference in Tfh cells CAMK4 expression between healthy donors and SLE patients (**Fig. S7A**). As previously shown²¹, CAMK4 expression is not affected by SLE disease activity (**Fig. S7B**), whereas its activation and nuclear localization (which mediates its genetic effect) is²¹."

Minor concerns:

1. In figure 2E, luciferase reporter vector was used to investigate the molecular mechanism of CaMK4 in Tfh cells. The authors should show the transfection efficiency. The details of nucleofection in primary T cells should be completed in the methods.

We are unable to show the transcription efficiency for the dual-luciferase reporter assay since the luciferase activity cannot be studied using cytometry. In our experiments, the luciferase signal was within 10^4 to 10^5 relative light units (RLU). The sensitivity of this assay (PROMEGA dual glo luciferase assay) is 10^2 - 10^6 RLU, which suggests that the transfection efficiency was good and our results within the margin of validity. Importantly, the fly luciferase activity (Bcl6 promoter vector activity) was normalized to a control renilla luciferase vector activity which was co-transfected with Bcl6 promoter vector and constantly express in cells. Therefore, transfection efficiency does not affect the results when the RLU are within the limits of detection. For the reviewer, we show below the transfection efficiency of the CaMK4-overexpression vector in another experiment, showing that our transfection protocol using electroporation is valid.

Figure (for the Reviewers only): murine CD62⁺CD4⁺ T cells were isolated and cultured in Tfh-polarizing conditions. At day 2 of cultures, the cells were transfected with 10 μ g of vector (empty vector [left panel] or CaMK4-overexpressing vector [right panel]). The results were studied at day 3.

In accordance with the reviewer's suggestion, we have completed the methods concerning the details of the nucleofector transfection in *in vitro* differentiated murine Tfh cells that were used in the study (pages 4-5, lines 174-186).

2. Numbers given for the amount of mice used per group or replicates of experiments should be added in Figure 1, 2, 3 and 6.

In accordance with the reviewer's suggestion, we have added the number of mice used for each experiment in Figures 1, 2, 3 and 6. As stated in the methods section, each point represents an individual mouse (*in vivo* experiment) or a biological replicate (*in vitro* experiment).

3. In figure 4C-D, the authors used two-way ANOVA for statistical analysis, it should be described in the Materials and Methods part.

We have described the two-way ANOVA analysis in the methods parts (see below).

Page 6, line 243-245: "To analyze the IgG response over time, we used a two-way ANOVA which take into account sampling time and the mice (matched values). Multiple testing was corrected using the Hold-Sidak's correction."

4. The gating strategy for all represented FACS plots should be added (eg. Figure2A, 5A).

As per the reviewer's suggestion, we added the gating strategy for the Fig. 2A (**Fig. S2A** of the revised manuscript) and Fig. 5A (in **Fig. S5A** of the revised manuscript).

Figure S2A: (A) Gating strategy of iTfh cells.

Figure S5A: (A) Gating strategy of Tfh in the *B6.lpr* mice spleen, cervical lymph nodes (cLNs) and peripheral blood.

5. Some results were less correlated with the whole manuscript (Figure 1D). It would be better to move these results in the supplementary figures to increase the logic and coherence

In accordance with the reviewer's suggestion, we have moved the Treg and Tfh results from the panel Fig. 1D to supplementary data (**Fig. S1G-H**).

Reviewer #2 (expert in TFH/B cell responses in health and disease, SLE):

This paper continues the investigations from the Tsokos group on the role of CAMK4 in T cell effector functions in relation to lupus pathogenesis, since CAMK4 is over expressed in lupus T cells. The present study showed that Camk4 is highly expressed in murine and human Tfh cells, and that CAMK4 directly binds to the Bcl6 promoter. Camk4-deficiency in T cells reduced Tfh development and subsequently that of GC B cells and humoral responses in responses to immunizations as well as in the *B6.lpr* model of autoimmunity. They also showed that purified human Tfh treated with a CAMK4 inhibitor had a reduced ability to help memory B cells to differentiate into plasma cells. These results add Tfh cells as an effector T cells that is regulated by CAMK4. This implies that the therapeutic effect of CAMK4 inhibition is multipronged and targets all the major T cell subsets that have been implicated in lupus pathogenesis.

The study is straightforward, and the results support the author's conclusions.

We appreciate the reviewer's kind assessment of our manuscript.

A few issues should be addressed:

What was the effect of CAMK4 deletion in T cells in *B6.Lpr* mice on Treg and Tfh

cells? Did it affect IFN γ and IL-17 production? Although the deletion in Tfh cells is most likely to be responsible for the B cell phenotypes, the decreases immune complex deposition in the kidneys could be attributed to other T cell subsets. The authors should discuss how the increased IL-2 production that they have previously reported following CAMK4 inhibition could also contribute to Tfh cell inhibition.

We thank the reviewer for raising these important points. We have previously shown that in the B6.*lpr* with a T-cell specific Camk4 deletion, there is an increase in splenic Treg cells, and a decrease of IL17-producing CD4⁺ T cells while IFN- γ remained unchanged (Scherlinger et al., Science Advances 2022).

We have added the data on the Treg cells on this set of mice, and found that splenic Treg cells were decreased whereas cLN Treg cells were not (Fig. S5I of the revised manuscript).

Figure S5I: Gating strategy (left panel), relative (middle panel) and absolute (right panel) of T regulatory (Treg) cell in the spleen and cervical lymph nodes (cLNs) of B6.*lpr.camk4^{fl/fl}* and B6.*lpr.camk4^{fl/fl}.dlckCre⁺* mice.

To address the reviewer's question about Tfr, we studied CXCR5⁺CD25⁺CD4⁺CD3⁺ Tfr cells in B6.*lpr* mice. We did not find differences in Tfr in B6.*lpr* mice Camk4-deficient T cells (Fig. S5C).

These results suggest that the effect of T-cell Camk4 deletion on the humoral autoimmune response is mainly driven by the suppression of Tfh cells, although increased splenic Treg cells may also play a role. We have added these results to the supplementary figures and discussed them in the revised manuscript as shown below.

Figure S5C: (C) gating strategy (left panel) and cumulative results (right panel) of T follicular regulatory cells in the *B6.lpr* model.

As suggested by the reviewer, we now discuss in the revised manuscript the potential role for increase IL-2 production in the inhibition of Tfh cell function and germinal center formation and added a new reference (35, André Ballesteros-Tato et al, Immunity 2012).

Page 10, lines 459-461: "For example, restored IL-2 production in CaMK4-deficient mice may participate to Tfh cell dysfunction and impaired follicular center formation³⁵."

CAMK4 expression is very low in B cells. Why was CAMK4 expression in human Tfh cells compared to that in B cells rather than other T cell subsets in both healthy subjects? The levels of CAMK4 expression in Tfh cells should be compared between HS and SLE patients.

We thank the reviewer for raising this point. We have now compared *CAMK4* expression in various T cell subsets and modified Figure 6 accordingly (see new **Fig. 6A** below).

Figure 6A: (A) T follicular helper (Tfh, CD3+CD4+CD25-CD127+CXCR5+) cells, T regulatory (Treg, CD3+CD4+CD25+CD127-CXCR5-) cells, T conventional (Tconv, CD3+CD4+CD25-CD127+CXCR5-) cells, B memory (CD3-CD19+CD27+IgD-) and B naïve (CD3-CD19+CD27-IgD+) cells were sorted from healthy donors peripheral blood mononuclear cells (n = 4 donors).

As suggested by the reviewer, and the first reviewer, we have compared *CAMK4* expression in healthy subject and SLE patients (**Fig. S7A**, below) and did not find a difference. These results are expected since we have previously shown that in patients with active SLE, the level of nuclear (active) CaMK4 protein compared to healthy control (Juang et al JCI 2005 & Scherlinger et al., Science Advances 2022).

Figure S7A: (A) Comparison of *CAMK4* expression using RT-qPCR in healthy donor (HD, $n = 7$) and SLE patients ($n = 13$) sorted primary Tfh cells.

Figure 4 from Juang et al, JCI 2005: Increased expression of CaMKIV in the nucleus of SLE T cells. (A) Both cytoplasmic and nuclear proteins were isolated from normal and SLE T cells. Western blots were then conducted by sequential blotting of the membrane with antibody against CaMKIV or CaMKII. Representative blots from 2 patient-control pairs are shown. (B) Eleven pairs were studied. Nuclear densitometric readings for CaMKIV, but not CaMKII, are higher ($P < 0.01$) in SLE T cells than in normal T cells. * $P < 0.05$

We have discussed this point in the revised manuscript.

page 9 lines 390-393: “There was no difference in Tfh cells *CAMK4* expression between healthy donors and SLE patients (**Fig. S7A**). As previously shown²¹, *CAMK4* expression is not affected by SLE disease activity (**Fig. S7B**), whereas its activation and nuclear localization (which mediates its genetic effect) is²¹.”

Minor
A common definition of ABCs is CD11c+ Tbet+

concerns:

We thank the reviewer for this point. We unfortunately did not conduct intracellular staining in the B cell panel used in our study. We used another definition of ABCs (B220⁺CD93⁻CD23⁻CD21/35⁻ cells) used by others (*ie*, Sisirak et al., Cell 2016 Jun 30;166(1):88-101).

Reviewer #3 (expert in calmodulin-dependent kinases):

In this manuscript, Scherlinger et al. investigated the role of CaMK4 expressed in T cells on the development of T follicular helper cells (Thf) and antibody response. To this purpose, the authors leveraged T-cell specific CaMK4 KO mice (Camk4 fl/fl DclK-Cre) combined with Lupus-prone mouse strains (B6.lpr). Human blood samples from SLE and healthy individuals were also used in the study.

The authors demonstrated that CaMK4 expressed in T cells is required to drive development of Thf cells and sustain B cells expansion in response to antigenic stimulation. Mechanistically, they propose that CaMK4 expressed in T cells is required to activate CREMa and drive BCL6 upregulation in Thf. Deletion of CaMK4 in T cells was also found associated with improved clinical features in lupus-prone mice. Data from human blood samples with KN93 drug inhibitor suggested that CaMK4 controls the expansion/development of in vitro generated Thf. Based on these results, the authors propose CaMK4-CREMa as an important signal pathway controlling T cell-dependent B cell response in autoimmunity.

The results included in this manuscript are interesting and identify a new function of CaMK4 in Thf cell biology and B cell response in normal and lupus-prone setting. However, previous data of showing the function of CaMK4 on T cell development, SLE, and immune-mediated disorders limit the novelty of this manuscript. Specifically, the role of CaMK4 in T cell subsets development (e.g., Th17 and Treg) alongside the ability of this protein to regulate critical functions of immune and non-immune cell types (e.g., myeloid cells, epithelial cells, endothelial cells, and podocytes), which have relevant roles in the pathogenesis of SLE and other immune-mediated/autoimmune diseases have been widely reported (e.g., Koga et al. 2012; Koga et al. JCI 2014; Ferretti et al. 2018; Maeda et al. 2018; Bhargava et al. 2021; Scherlinger et al. 2022; Yong et al. 2022). Similarly, the ability of CaMK4 to modulate CREMa (and CREB) activity in T cells has been already previously documented (Koga et al. JCI 2014;). Lastly, the role of CaMK4 in the modulation of B cell response to vaccine has been previously investigated (Nakaya et al. 2012), while the discrepancies between the results from this earlier publication with data included in the current manuscript have not been sufficiently discussed/investigated. In conclusion, the finding that CaMK4 regulates Thf functions is interesting and novel, while at least in part, this study provides incremental pieces of evidence supporting the well-known role of CaMK4 in SLE/autoimmunity.

We thank the Reviewer for his thorough evaluation which led to his positive appraisal of our work. Please see below detailed response to the reviewer's comments and questions.

Major criticisms

1. The absolute cell count number should be provided and shown in all datasets/figures. This is an important finding to clarify/distinguish between direct and non-direct effects on cell subsets.

In accordance with the reviewer's suggestion, we have added the absolute cell count through the revised manuscript and figures.

2. The complete immune profile of mice used in this study (basal and after antigenic stimulation) should also be provided.

We thank the reviewer for this raising this point. We have added in the revised manuscript a detailed immune cells characterization of $Camk4^{fl/fl}$ and $Camk4^{fl/fl}.dlck^{Cre}$ mice:

- T follicular helper cell pre- and post-antigenic stimulation (**Fig. 1A**; and **Fig S1A**)
- T peripheral helper cell pre- and post-antigenic stimulation (Fig. **S1B-C**)
- T regulatory cell and T follicular regulatory cells pre- and post-antigenic stimulation (**Fig. S1G-H**)
- B cell subsets including age-related B (ABC) cells and follicular B (FOB) cells pre- and post-antigenic stimulation (**Fig. S3B-D**).
- Germinal center (GC) B cells pre- and post-antigenic stimulation (**Fig. 3A** and **Fig. S3A**)

For all these data, both relative and absolute number of cells are reported, as per the reviewer's recommendation (see below).

Figure 1B: (B-D) $Camk4^{fl/fl}$ (red, $n = 5$) and $Camk4^{fl/fl}.dlck-Cre$ (blue, $n = 5$) mice were immunized by an i.p. injection of sheep red blood cells and subsequently sacrificed after 7 days. Non-immunized mice were used as a control ($n = 4$ for each genotype). (B) Gating strategy of $PD1^+$ T_{fh} cells and $CXCR5^+PD1^+$ T peripheral helper (T_{ph}) cells from the spleen of $Camk4^{fl/fl}$ (red) and $Camk4^{fl/fl}.dlck-Cre$ (blue) 7 days after immunization with sheep red blood cells (left panel) and cumulative results (right panel).

Figure S1: Supplementary results in B6.Camk4^{fl/fl}.dlck^{Cre} mice immunized with sheep red blood cells. (A) Absolute number of splenic PD1⁺ Tfh cells. (B-C) Relative (B) and absolute (C) number of splenic PD1⁺ CXCR5⁺ T peripheral helper (Tph) cells. (D) Absolute number of splenic Bcl6⁺ Tfh cells. (E-F) Relative (E) and absolute (F) number of splenic PD1⁺ Tfh cells in wild-type (C57Bl/6) and *Camk4*^{-/-} mice immunized with SRBC (day 7). (G) Gating strategy (left panel) and relative number (right panel) of T regulatory (Treg) cells at the basal state and after immunization. (H) Absolute number (right panel) of T regulatory (Treg) cells at the basal state and after immunization. Each point represents one mouse, bars indicate mean \pm s.e.m. *, $p < 0.05$; **, $p < 0.01$; *, $p < 0.001$; ****, $p < 0.0001$ using one-way ANOVA with Holm-Sidak's correction.**

Figure S3: (A) Absolute numbers of splenic germinal center (GC) B cells in *Camk4^{fl/fl}* (red) and *Camk4^{fl/fl}.dlck^{Cre}* (blue) mice immunized with sheep red blood cells (SRBC). (B) Gating strategy of B cells subtypes in the SRBC immunization model. (C) Relative number of B cell subtypes in the SRBC immunization model. (D) Absolute numbers of splenic B cell subtypes. Each point represents one mouse, bars indicate mean \pm s.e.m. *****, $p < 0.0001$ using one-way ANOVA with Holm-Sidak's correction.

3. KN93 is a nonspecific inhibitor of CaMK4 and this compound blocks additional relevant targets in T cells (e.g., CaMKII). The author should address the specificity of this compound in T cells by comparing the effects of KN93 treatment on lymphocytes isolated from WT and CaMK4-deficient mice.

We agree with the reviewer that the kinase inhibitor KN93 is a not entirely specific of CaMK4. For this reason, we used genetic deletion of the *Camk4* gene in all mice studies conducted in the manuscript. We used KN93 only in experiment using human cells where genetic deletion is challenging. To address the reviewer's point of the specificity of CaMK4 inhibition, we instead conducted a silencing of *CAMK4* using siRNA in human T cell to confirm that KN93 effect was mediated by CaMK4 inhibition. Please see the experiment below that was suggested by the reviewer.

4. A gene silencing approach should be used to assess the function of CaMK4 on human development of Thf cells.

To address this point, we sorted human CD4⁺CD25⁻CD127⁺CXCR5⁺ T cells from healthy donors (n = 4 donors). We transfected these cells with *CAMK4* target or control siRNA and evaluated BCL6 gene expression after 48 hours. We found that *CAMK4* target siRNA decreased significantly *CAMK4* expression. *CAMK4* silencing was responsible for a significant downregulation of BCL6 expression in human Tfh cells. We have added these new data in the revised manuscript.

Figure S6: CAMK4 silencing in human primary Tfh cells results in BCL6 downregulation. Human primary CD4⁺CD25⁺CD127⁺CXCR5⁺ Tfh cells were sorted from healthy donors using FACS (n = 4). The cells were transfected with control or CAMK4-target siRNA. Two days after transfection, RNA was extracted, and qPCR conducted. Relative expression of CAMK4 (A) and BCL6 (B). Each dot indicates an independent experiment from a healthy donor, bar show mean \pm s.e.m. *, $p < 0.05$; ****, $p < 0.0001$ using paired Student's t-test.

5. Germline global deletion of CaMK4 has been associated with enhanced response to inactivated flu trivalent vaccine (Nakaya et al. Nat Immunol, 2012). On the opposite, data included in the present manuscript demonstrated that Camk4 fl/fl Dclk-Cre mice secrete lower amount of antibody in response to SRBC and NP antigenic stimulation, compared to WT mice. Discrepancies between global and T cell-specific KO should be discussed or eventually addressed by comparing the effects of germline global CaMK4 KO and T cell specific deletion on antigenic stimulation, including flu trivalent inactivated vaccine.

We thank the reviewer for raising this important point. Similarly, to mice with T-cell specific *Camk4* deletion, global *Camk4* knockout mice demonstrate impaired Tfh cell expansion upon SRBC immunization (Fig. S1E-F), and impaired high affinity antibody response to NP-CGG (Fig. S4D).

Figure S1E-F: (E-F) Relative (E) and absolute (F) number of splenic PD1⁺ Tfh cells in wild-type (C57Bl/6) and Camk4^{-/-} mice immunized with SRBC (day 7). **, $p < 0.01$; ****, $p < 0.0001$ using one way ANOVA with Holm-Sidak's correction.

Figure S4D: Wild-type ($n = 5$ mice) and *Camk4*^{-/-} ($n = 3$ mice) were immunized with NP-CGG with a recall injection at day 30. High affinity (anti-NP-7, left panel) and mixed affinity (anti-NP-44, middle panel) IgG were measured at different time points. The ratio of NP-7/NP-44 IgG was calculated at the last time point (7 days after recall immunization). *, $p < 0.05$; **, $p < 0.01$ using two-way ANOVA (left and middle panel) or unpaired Student's t-test (right panel).

The differences with *Nakaya et al.* might be explained by the different route of immunization (intraperitoneal vs intramuscular injections) or the presence of adjuvant (Alum in our study vs no adjuvant). Indeed, different vaccination route results in different diffusion of the antigen to secondary lymphoid organs that might impact the T/B cell response to the antigen. While we believe immunization with flu trivalent inactivated vaccine is outside the scope of our work, we have discussed this point in the discussion part of the revised manuscript.

Page 10, lines 420-426: "Nakaya and colleagues²³ also showed that mice with a *Camk4* germline deletion developed enhanced IgG response to flu vaccines, which is in contrast to our results with NP-CGG immunization (**Fig. S4D**). The differences might be explained by the different route of immunization (intraperitoneal vs. intramuscular injections)²⁵ or the presence of adjuvant (Alum in our study vs. no adjuvant). Indeed, different vaccination route results in different diffusion of the antigen to secondary lymphoid organs that might impact the T/B cell response to the antigen."

6. The title should be revised to reflect more closely the results of the study. For example, it should be clearly stated that the study focusses on the role of CaMK4 expressed in T cells on development and antibody response in normal and SLE setting. A more general statement of the role of CaMK4 on autoimmunity would imply the assessment of global and cell-specific (e.g., T cell, myeloid cell) deletion of CaMK4 in normal/pathological antibody response, including additional autoimmune models.

In accordance with the Reviewer's point, which was also raised by Reviewer #1, we have modified the title as follow:

"CaMK4 controls Tfh expansion and function during normal and autoimmune T-dependent B cell response"

Minors:

Figure

a) Panel C: CXCR5⁻ and CXCR5⁺ seem to express different levels of BCL6. However,

1.

the gating strategy does not take into consideration this possible relevant difference. Gate should focus on BCL6^{high} and results would include % and absolute cell number. Alternatively, BCL6 gene/protein expression in CXCR5- and CXCR5+ cells should be assessed with additional methods (e.g. qPCR, immunoblot) using purified cells.

We have added the absolute cell number that refers to Fig. 1C (Fig. S1D in the revised manuscript).

In accordance with the reviewer's suggestion, we show another Tfh cell gating that focuses on BCL6^{hi} cells. Focusing on Bcl6^{hi} cells does not alter the results, that is mice which do not express CaMK4 in T cells have reduced splenic Bcl6^{hi} Tfh cells. We did not add this analysis to the revised manuscript as it does not change the conclusion.

Figure (for the reviewer only): (A) Gating strategy of splenic CD4⁺ CXCR5⁺ Bcl6^{hi} T_{fh} cells (left panel) and cumulative results (right panel). (B) Absolute number of splenic CD4⁺ CXCR5⁺ Bcl6^{hi} T_{fh} cells.

Figure 2:

1. Panel B: Please, clarify whether the expression of BCL-6 (MFI) refers to total CD4 T cells.

The MFIs refers to the PD1⁺ Bcl6⁺ Tfh cells as gated in Figure 2A. We have changed the graph to indicate that the MFI refers to these cells, as well as in the legend.

2. Panel C: Not cut gel should be provided, CAMK4 expression should be shown. Actin expression between WT and KO is significantly lower. Please, explain and eventually normalize BCL-6 expression using an additional/alternative housekeeping gene (e.g., Tubulin). Please, clarify if dots refer to technical or biological (from independent experiments) replicates.

As requested by the reviewer, we provide the uncut gels, as supplementary files for review. Furthermore, we have added the staining for CaMK4 protein to confirm that the bands on the right come from *Camk4*^{-/-} mice.

Sometimes, the protein content is different between samples which results in lower intensity of bands. However, we used the β -actin as control and we normalized the Bcl6 expression using β -actin.

As indicated in the statistical analysis in the methods section, unless stated otherwise all plotted points represent an independent experiment/biological replicates. We have added this point in the legend of all figures.

Figure

3

1. Panel B: quantitative data on GC should be provided and shown in the figure.

This point was also raised by Reviewer 1. To address the reviewer's comment, we have quantified the areas of GC in $Camk4^{fl/fl}$ and $Camk4^{fl/fl}.dlck^{Cre}$ mice ($n = 3$ per group; figure 3D). We have added this data in the revised manuscript (Fig 3D-E).

Figure 3C-D legend: (C) Immunofluorescence of the spleen 7 days after immunization showing T cell (CD3, green), B cells (B220, red) and germinal centers (peanut antigen PNA, blue). (D) The area of germinal centers (PNA staining) was measured in the spleen of $Camk4^{fl/fl}$ and $Camk4^{fl/fl}.dlck^{Cre}$ ($n = 3$ per group). Each dots represents the area of one germinal center. ***, $p < 0.001$ using Kolmogorov-Smirnov test.

Figure 5

1. Assessment of T cell subsets (Naïve, CM, EM, Treg, and Tfh), which include % and absolute cell number should be provided.

In accordance with the reviewer's suggestion, we have added these data in the revised manuscript (Fig. S5A and Fig. S5H-I).

Figure 5: (H) Splenic CD4⁺ T cell subsets were studied using cytometry. (I) T regulatory (Treg) cells were evaluated in the spleen and cLN. *, $p < 0.05$; **, $p < 0.01$ using one way ANOVA with Holm-Sidak's correction.

2. Panel H-I: green staining cannot be visible. Please, provide better images and clarify how Ig and C3 signals have been quantified. We have provided new images and added in the methods section how the signals were quantified.

Figure

6:

1. Panel A: specify which housekeeping gene has been used for normalization. The housekeeping gene used for normalization is GADPH. This information has been added to the legend, and the reference of the probe used (ref. Hs02758991_m1) can be found in Table S3.

2. Panel B and D: Provide absolute cell count numbers.

Panel 6B represents a qPCR, we are not sure what the reviewer means.

Panel D represents cytometry results from an in vitro experiment using human cells. We have calculated the absolute number of plasmablast cells since our cytometer (Cytotflex) allows us to count cells. Significant decrease was confirmed using absolute number of cells (see figure below for the reviewer).

Figure (for the reviewer): absolute numbers of CD38⁺CD27⁺ B cells at the end of the coculture experiment.

Discussion, 337-344: This paragraph should be revised, including more recent evidence.

Specifically, the authors mentioned that results from Nakaya study (Nakaya et al. 2012), showing a negative correlation between CaMK4 expression in PBMC at day 3 post-flu-vaccination and antibody response at day 28, may depend on B cell expansion and the corresponding dilution of T cells specific genes, which should include CaMK4. Although qualitatively correct, a more careful quantitative analysis of B cell expansion in vaccinated individuals (approximately 10e3 B cells per million of PBMC) would not provide a robust/reasonable support to this hypothesis. The authors also mentioned that a more recent extensive analysis, which analyzed the response against 13 vaccines in 3000 individuals (Hagan et al. 2022), does not confirm the original finding on CaMK4 downregulation in high-responder individuals. However, the authors should also mention that a major conclusion of that large study is the failure to identify a “universal” predictive signature between the different vaccines. On the other hand, CaMK4 expression was also not found positively associated with an enhanced B cell response, as predicted by the results of the present manuscript. Lastly, the negative correlation between CaMK4 expression and antibody titer has been recently documented in a recent study assessing the influenza A/H3N2 antibody response to high-dose influenza and adjuvanted influenza vaccine in older adults (Haralambieva et al. 2022). This study should be mentioned and discussed.

Following the reviewer's comment, we have removed the mention of B cell expansion to try to explain the results of Nakaya et al study. We have also mentioned that Hagan et al. failed to identify any signature gene for good vaccine response. After reading carefully the Haralambieva paper mentioned by the reviewer, we noticed that after correcting for multiple analysis (transcriptomics analysis), the false discovery rate (FDR-) q value of *Camk4* is 0.1575, a result that is not significant. Furthermore, the authors found that CaMK4 expression was positively associated with A/H3N2 antibody response from the Fluzone quadrivalent vaccine, although this was not statistically significant.

To address the reviewer's point, we have discussed these results in the revised manuscript (see below).

Page 9-10 lines 413-426: "Interestingly, our results are different to those observed by Nakaya and coauthors²³. They observed that the day 3 expression of *CAMK4* in PBMCs from individuals immunized with the inactivated influenza vaccine negatively correlated with the vaccinal antibody response at day 28²³. However, a recent transcriptomic atlas encompassing more than 3000 individuals immunized with 13 different vaccines failed to identify an association between *CAMK4* expression (or any other gene) and antibody response²⁴. Nakaya and colleagues²³ also showed that mice with a *Camk4* germline deletion developed enhanced IgG response to flu vaccines, which is in contrast to our results with NP-CGG immunization (**Fig. S4D**). The differences might be explained by the different route of immunization (intraperitoneal vs. intramuscular injections)²⁵ or the presence of adjuvant (Alum in our study vs. no adjuvant). Indeed, different vaccination route results in different diffusion of the antigen to secondary lymphoid organs that might impact the T/B cell response to the antigen.

"

REVIEWERS' COMMENTS

Reviewer #1 (expert in genomics of SLE and therapeutic targets for SLE):

The response in the revision is commendable and very thorough. All necessary supplementary experiments have been conducted, and it successfully addresses all the concerns raised.

Reviewer #2 (expert in TFH/B cell responses in health and disease, SLE):

all concerns have been thoroughly addressed and the new results have strengthened the paper.

Reviewer #3 (expert in calmodulin-dependent kinases):

The authors have consistently addressed all major concerns. However, a few minor issues still need to be addressed.

1. The quality of the uncut CaMK4 gel is marginally acceptable. However, it becomes more challenging to identify the CaMK4 band due to the absence of a protein molecular marker ladder. Please consider adding the molecular weight markers to the uncut gel for better clarity. I also recommend including this figure as supplementary data. This will help readers/colleagues to fully evaluate the datasets included in the main text and help to design/interpret similar studies in the field.

2. In their rebuttal, the authors discussed the discrepancies between the data from Nakaya et al. (ref. 23) and more recent publications, which focused on a secondary meta-analysis using transcriptomics data available in a public repository (ref 24). Given the complexity and heterogeneity of the human population, datasets, and statistical approaches, it is not surprising that this latter study failed to identify CaMK4 or any other genes as biomarkers associated with vaccine outcomes. On the other hand, Haralambieva et al. recently included CaMK4 in the list of genes associated with the flu vaccine response ($p < 0.001$). However, the False Discovery Rate (FDR) for this and other genes included in the list by these Authors was not significant. Again, heterogeneity in individual responses, sample collection and preparation, and data analysis may impinge on the FDR analysis. Nonetheless, I believe that the results from Haralambieva's work provide a valuable contribution to addressing the discrepancies in the field and should be discussed and included in the manuscript.

RESPONSE TO REVIEWERS' COMMENTS

Reviewer #1 (expert in genomics of SLE and therapeutic targets for SLE):

The response in the revision is commendable and very thorough. All necessary supplementary experiments have been conducted, and it successfully addresses all the concerns raised.

We thank the Reviewer for his/her positive assessment of our revised manuscript.

Reviewer #2 (expert in TFH/B cell responses in health and disease, SLE):

all concerns have been thoroughly addressed and the new results have strengthened the paper.

We thank the Reviewer for his/her positive assessment of our revised manuscript.

Reviewer #3 (expert in calmodulin-dependent kinases):

The authors have consistently addressed all major concerns. However, a few minor issues still need to be addressed.

We thank the Reviewer for his/her positive assessment of our revised manuscript.

1. The quality of the uncut CaMK4 gel is marginally acceptable. However, it becomes more challenging to identify the CaMK4 band due to the absence of a protein molecular marker ladder. Please consider adding the molecular weight markers to the uncut gel for better clarity. I also recommend including this figure as supplementary data. This will help readers/colleagues to fully evaluate the datasets included in the main text and help to design/interpret similar studies in the field.

The molecular weight marker is on the original membrane but difficult to read because overexposed. To address this we have added labelled the molecular weight on the figures and added to supplementary data (Fig. S8) as per the Reviewer recommendation.

2. In their rebuttal, the authors discussed the discrepancies between the data from Nakaya et al. (ref. 23) and more recent publications, which focused on a secondary meta-analysis using transcriptomics data available in a public repository (ref 24). Given the complexity and heterogeneity of the human population, datasets, and statistical approaches, it is not surprising that this latter study failed to identify CaMK4 or any other genes as biomarkers associated with vaccine outcomes. On the other hand, Haralambieva et al. recently included CaMK4 in the list of genes associated with the flu vaccine response ($p < 0.001$). However, the False Discovery Rate (FDR) for this and other genes included in the list by these Authors was not significant. Again, heterogeneity in individual responses, sample collection and

preparation, and data analysis may impinge on the FDR analysis. Nonetheless, I believe that the results from Haralambieva's work provide a valuable contribution to addressing the discrepancies in the field and should be discussed and included in the manuscript.

To address Reviewer request, we now cite the study by Haralambieva and colleagues (Reference 24) in the revised manuscript.

“Interestingly, our results are different to those observed by Nakaya and coauthors²². They observed that the day 3 expression of *CAMK4* in PBMCs from individuals immunized with the inactivated influenza vaccine negatively correlated with the vaccinal antibody response at day 28²². However, a recent transcriptomic atlas encompassing more than 3000 individuals immunized with 13 different vaccines failed to identify an association between *CAMK4* expression (or any other gene) and antibody response²³. Furthermore, Haralambieva and colleagues found that T-cell *CAMK4* expression was (non-significantly) negatively associated with the Flud vaccine antibody response and (non-significantly) positively associated with the Fluzone vaccine antibody response²⁴. Nakaya and colleagues²² also showed that mice with a *Camk4* germline deletion developed enhanced IgG response to flu vaccines, which is in contrast to our results with NP-CGG immunization (**Fig. S4D**). The differences might be explained by the different route of immunization (intraperitoneal vs. intramuscular injections)²⁵ or the presence of adjuvant (Alum in our study vs. no adjuvant). Indeed, different vaccination route results in different diffusion of the antigen to secondary lymphoid organs that might impact the T/B cell response to the antigen. “